# Both maternal IFNγ exposure and acute prenatal infection with *Toxoplasma gondii* activate fetal hematopoietic stem cells

Diego A López[1,†] , Kelly S Otsuka[1,†] , April C Apostol[2,†] , Jasmine Posada[3] , Juan C Sánchez-Arcila[3] ,
Kirk DC Jensen[3,4] & Anna E Beaudin[5,*]

## Abstract

Infection directly influences adult hematopoietic stem cell (HSC) function and differentiation, but the fetal hematopoietic response to infection during pregnancy is not well-studied. Here, we investigated the fetal hematopoietic response to maternal infection with *Toxoplasma gondii* (*T. gondii*), an intracellular parasite that elicits Type II IFNγ-mediated maternal immunity. While it is known that maternal infection without direct pathogen transmission can affect fetal immune development, the effects of maternal IFNγ on developing HSCs and the signals that mediate these interactions have not been investigated. Our investigation reveals that the fetal HSCs respond to *T. gondii* infection with virulence-dependent changes in proliferation, self-renewal potential, and lineage output. Furthermore, maternal IFNγ crosses the fetal–maternal interface, where it is perceived by fetal HSCs. By comparing the effects of maternal IFNγ injection with maternal *T. gondii* infection, we reveal that the effects of IFNγ treatment mimic some aspects of the fetal HSC response to infection. Moreover, our findings illuminate that the fetal HSC response to prenatal infection is distinct from the adult HSC response to IFNγ-induced inflammation. Altogether, our data disentangle the role of infection-induced inflammatory cytokines in driving the expansion of downstream hematopoietic progenitors.

**Keywords** congenital infection; hematopoiesis; hematopoietic stem cell; inflammation; *Toxoplasma gondii*
**Subject Categories** Development; Microbiology, Virology & Host Pathogen Interaction
**The EMBO Journal (2023) 42: e112693**

## Introduction

Congenital infection can have dire outcomes for fetal health and development. Several pathogens are implicated in vertical transmission or infection of the fetus from the maternal host. These so called "TORCH" pathogens include *Toxoplasma gondii*, "other" (syphilis, varicella-zoster, parvovirus B19) (Stegmann & Carey, 2002), Rubella virus, Cytomegalovirus, and Herpes simplex virus (Megli & Coyne, 2022) and have recently been extended to include Zika virus (Kovacs, 2020). Even in the absence of vertical transmission, prenatal inflammation during infection can cause systemic changes in fetal immunity, including systemic cytokine production and lymphocyte polarization (Dauby *et al*, 2012; Levy & Wynn, 2014). These changes in fetal immunity translate into alterations in neonatal functional immune outcomes, including altered response to vaccine and susceptibility to infection (Apostol *et al*, 2020). Several proposed mechanisms may underlie this phenomenon, including the transplacental transport of maternal antibody–antigen complexes that directly prime fetal immune cells, or the active or passive transplacental transport of maternal inflammatory mediators or metabolites that directly stimulate an immune response in the fetus (Apostol *et al*, 2020). However, very little is known about how these signals are "seen" and "translated" by fetal hematopoietic and immune cells.

With respect to the adult hematopoietic system, accumulating evidence from studies on the effects of inflammation and infection indicate that these events modulate hematopoietic output by directly influencing hematopoietic stem cell (HSC) function (Pietras, 2017; Caiado *et al*, 2021). Inflammation can directly activate HSCs residing in the bone marrow (BM) due to their capacity to respond to inflammatory stimuli, derived from either circulation (Baldridge *et al*, 2011) or the BM niche (Mitroulis *et al*, 2020). Hematopoietic stem cell have the capacity to respond directly to a variety of inflammatory cytokines including TNF-α (Yamashita & Passegué, 2019), IL-1β (Pietras *et al*, 2016), IL-27 (Furusawa *et al*, 2016), and interferons

1 Division of Microbiology and Immunology, Department of Pathology, University of Utah School of Medicine, Salt Lake City, UT, USA
2 Quantitative and Systems Biology Graduate Program, University of California, Merced, Merced, CA, USA
3 Department of Molecular and Cell Biology, University of California, Merced, Merced, CA, USA
4 Health Science Research Institute, University of California, Merced, Merced, CA, USA
5 Departments of Internal Medicine and Pathology, and Program in Molecular Medicine, University of Utah School of Medicine, Salt Lake City, UT, USA
*Corresponding author. Tel: +1 801 581 2036; E-mail: anna.beaudin@hsc.utah.edu
†These authors contributed equally to this work

(Essers *et al*, 2009; Baldridge *et al*, 2010; Matatall *et al*, 2014; Haas *et al*, 2015). In addition to direct "sensing" of cytokines, adult HSCs also respond directly to signals from bacterial (Matatall *et al*, 2016) and viral (Hirche *et al*, 2017) infections. Across a wide variety of inflammatory stimuli, BM HSCs stereotypically respond by restricting lymphoid cell production in favor of myeloid expansion, mediated by activation of myeloid-biased HSCs and expansion of myeloid-primed progenitors (MacNamara *et al*, 2011b; Matatall *et al*, 2014; Haas *et al*, 2015; Pietras *et al*, 2016). Inflammation also causes adult HSCs to rapidly exit quiescence, which has negative consequences for their ability to self-renew and persist following transplantation (Essers *et al*, 2009; Baldridge *et al*, 2010; MacNamara *et al*, 2011a; Matatall *et al*, 2016; Hirche *et al*, 2017).

There is little information about the impact of prenatal inflammation on developmental hematopoiesis during early life. Proinflammatory cytokine signaling is required for normal pregnancy and fetal development but is also detrimental to fetal health during maternal infection (Yockey & Iwasaki, 2018). In early development, "sterile" inflammatory cytokine signaling is necessary for proper HSC specification in the ventral aortic endothelium (Espin-Palazon *et al*, 2018); both Type I and II interferon signaling regulate HSC specification (Li *et al*, 2014; Sawamiphak *et al*, 2014), as the lack of signaling contributes to fewer fetal hematopoietic stem and progenitor cells (HSPCs) overall. Emerging evidence suggests that fetal HSPCs, including HSCs, may also be amenable to inflammatory signals perceived from maternal sources (Apostol *et al*, 2020) and our laboratory recently established that fetal HSPCs respond directly to prenatal inflammation. Using a mouse model of maternal immune activation (MIA) induced by poly(I:C), we demonstrated that Type I interferons induced by prenatal inflammation activate transient, lymphoid-biased hematopoietic progenitors, driving a lymphoid-biased response during fetal hematopoiesis (López *et al*, 2022).

To gain further insight into the mechanism by which prenatal infection and inflammation affects fetal HSC development, here, we utilized a novel model of maternal infection using the intracellular apicomplexan parasite, *Toxoplasma gondii* (*T. gondii*). As a vertically transmitted and ubiquitous pathogen (Megli & Coyne, 2022), acute maternal infection is estimated to occur in approximately 1% of all human pregnancies (Rostami *et al*, 2019; Bigna *et al*, 2020) with roughly 10% of exposures resulting in fetal infection (Torgerson & Mastroiacovo, 2013). Congenital toxoplasmosis is recognized as an impediment to proper fetal and neonatal development (McLeod *et al*, 2009) with complications ranging from miscarriage, stillbirth, fetal death, neurological sequalae, chorio-retinitis, and hydrocephalus. *T. gondii* infections of human placentas *ex vivo* reveal certain cell types are better suited to restrict infection, and specific chemokine responses have been identified (Robbins *et al*, 2012; Ander *et al*, 2018). In mouse models of congenital infection, Type II IFNγ-mediated maternal immunity simultaneously promotes parasite clearance and prevents vertical transmission; however, the production of excessive IFNγ from *T. gondii* infection can also have dire consequences for the developing fetus, such as stillbirth and spontaneous abortion (Shiono *et al*, 2007; Pappas *et al*, 2009; Senegas *et al*, 2009). Beyond the well-established adverse outcomes associated with *T. gondii* congenital infection and identification of certain host resistance mechanisms, it is less clear how maternal infection impacts the fetal HSC compartment and subsequent immune system development.

To investigate the effect of congenital toxoplasmosis on developing fetal HSCs, we performed experiments using the "FlkSwitch" fate-mapping model, which identifies two functionally distinct fetal HSC populations: a lymphoid-biased, developmentally restricted HSC (drHSC) marked by Flk2-driven GFP expression and a "canonical" HSC marked by Tomato expression that persists into the adult bone marrow (Beaudin *et al*, 2016). We have recently demonstrated that the drHSC is sensitive to developmental perturbation with prenatal inflammation, and the activation of the drHSC by prenatal inflammation has direct consequences for postnatal immune and hematopoietic outcomes (López *et al*, 2022). To investigate how IFNγ directly mediates the fetal hematopoietic response within the complex framework of infection, we also compared the effects of maternal *T. gondii* infection with varying degrees of virulence to a single maternal cytokine injection of IFNγ. Our results demonstrate, in both an infection and Type II interferon model, that fetal HSCs are directly responsive to maternally-derived cytokines. Maternal infection also drives an independent inflammatory response within the fetus that evokes functional changes in HSCs and downstream multipotent progenitors. This study reveals the intricate response of fetal hematopoiesis to a *bona fide* congenital infection and positions IFNγ as a critical regulator of fetal HSC function. Our data suggest that individuals exposed to maternal *T. gondii* infection or Type II IFNγ-mediated inflammation *in utero* may exhibit functional alterations to their immune systems driven by changes to fetal HSC function.

# Results

### *In utero* exposure to *Toxoplasma gondii* impacts fetal hematopoiesis

To determine the direct effects of maternal infection on fetal hematopoiesis, we employed a mouse model of maternal infection using the "TORCH" or congenitally transmitted pathogen *Toxoplasma gondii* (*T. gondii*), a ubiquitous parasite that elicits a well-characterized IFNγ-mediated immune response during pregnancy (Shiono *et al*, 2007; Senegas *et al*, 2009). At embryonic day (E) 10.5, we injected pregnant dams with $2 \times 10^4$ tachyzoites (Fig 1A) of either the Pru or RH strains of *T. gondii*. The Pru strain is of "intermediate" virulence with a $LD_{50}$ of $2 \times 10^3$ parasites, whereas RH is considered highly virulent with an $LD_{100}$ of one parasite (Saeij *et al*, 2005). This dichotomy in virulence is evident in fetal size at E16.5, as increased virulence is associated with lowered crown–rump length (Figs 1B and EV1A) and decreased fetal viability (Fig EV1B).

To investigate how specific populations within the fetal liver (FL) hematopoietic stem and progenitor cell (HSPC) compartment responded to maternal *T. gondii* infection, we used previously described surface markers to demarcate long-term (LT-) and short-term (ST-) hematopoietic stem cells (HSCs) along with multipotent progenitor (MPP) 2, 3, and 4 subsets and Tom+ HSCs and GFP+ developmentally-restricted (dr) HSCs (Figs 1C–E and EV1C and D) (Yilmaz *et al*, 2006; Beaudin *et al*, 2016). Maternal *T. gondii* infection significantly affected fetal hematopoiesis at E16.5 in a virulence-dependent manner. First, in response to maternal infection, the frequency (Fig 1C–E) of HSPCs was increased in response

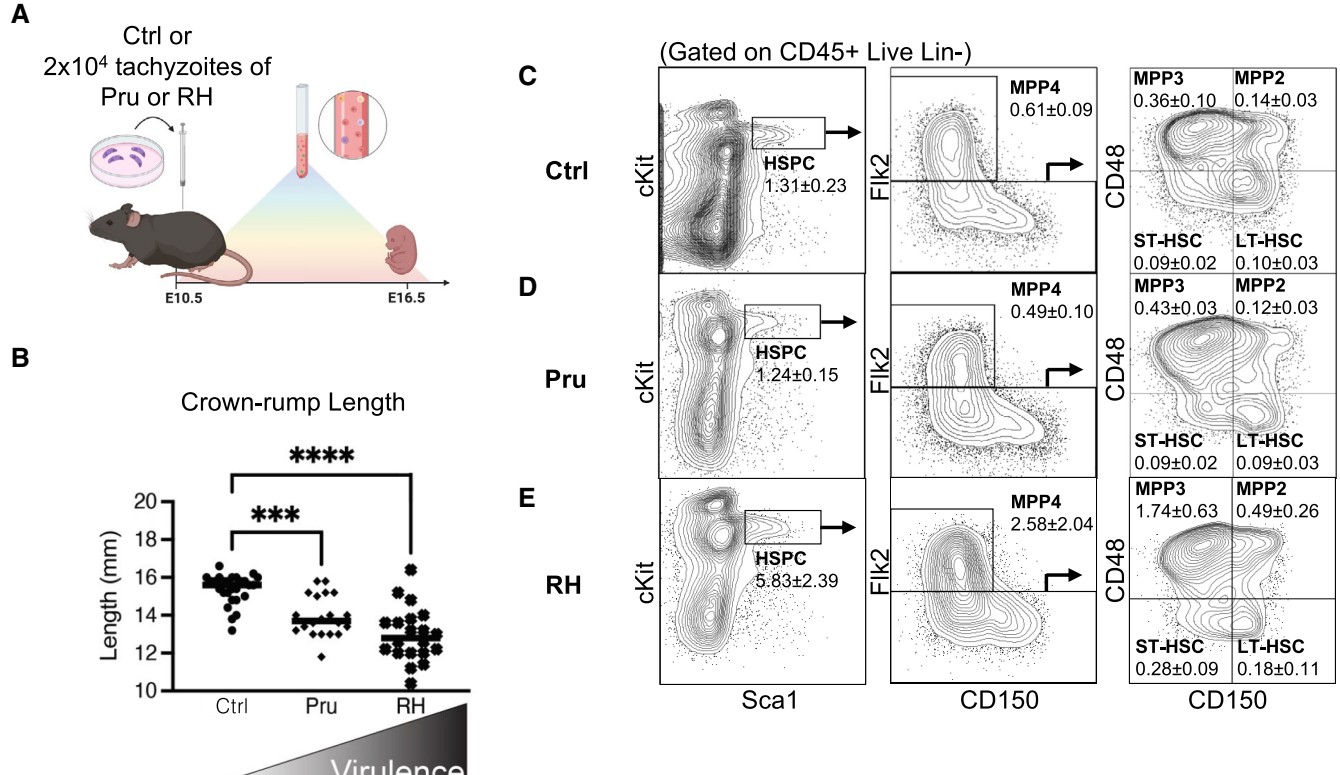

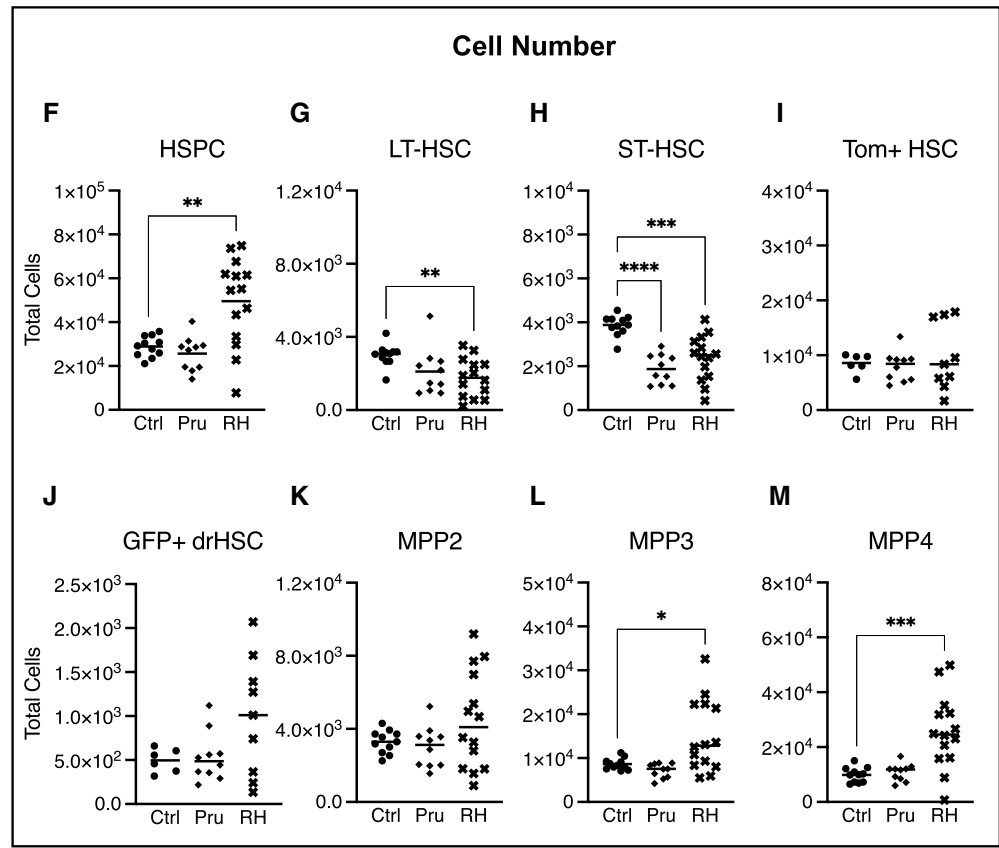

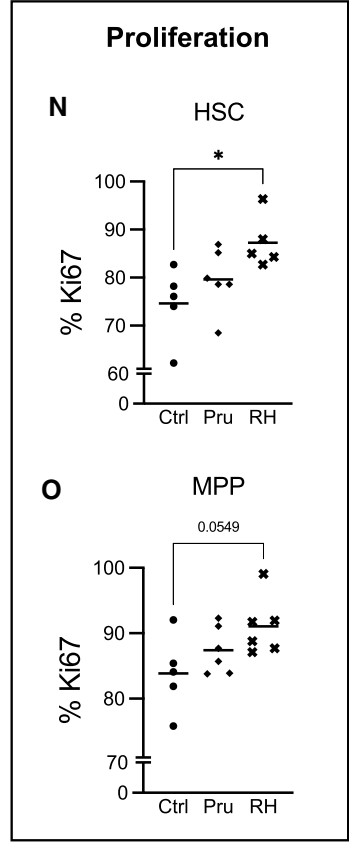

**Figure 1.**

**Figure 1.  In utero exposure to Toxoplasma gondii impacts fetal hematopoiesis.**

A    Schematic of prenatal infection with *Toxoplasma gondii*. At E10.5, pregnant FlkSwitch dams were injected with saline or $2 \times 10^4$ tachyzoites of parasite strains Pru or RH. Fetal outcomes were assessed at E16.5.

B    Crown–rump length (fetal size) was measured in E16.5 fetuses following saline or maternal infection with Pru or RH strains of *Toxoplasma gondii*.

C–E  Representative gating strategy and frequencies of various hematopoietic stem and progenitor cell (HSPC) populations in E16.5 fetal liver after (C) saline, (D) Pru, or (E) RH conditions. Numbers indicate frequency $\pm$ SD as a percentage of total CD45+, live, Lin− cells.

F–M  Total cellularity of (F) HSPCs, (G) LT-HSCs, (H) ST-HSCs, (I) Tom+ HSCs, (J) GFP+ drHSCs, (K) MPP2, (L) MPP3, (M) MPP4 in E16.5 fetuses following saline or maternal infection with Pru or RH as shown in (A). $n = 9$–15 fetuses from at least 3 litters/condition.

N, O  Frequency of (N) HSCs (CD150$^{hi}$) (O) MPPs (CD150− HSPCs) expressing Ki67 (G1+G2-M-S) at E16.5 following saline or maternal infection with Pru or RH as shown in (A). $n = 9$–15 fetuses from at least 3 litters/condition.

Data information: For all analysis above bars represent mean $\pm$ SEM. One-way ANOVA with Tukey's test. *$P \leq 0.05$; **$P \leq 0.01$; ***$P \leq 0.001$; ****$P \leq 0.0001$.
Source data are available online for this figure.

to the more virulent RH infection but unchanged in the Pru condition. The increased frequency was observed across all RH-exposed HSPCs (Figs 1E and EV1E–L). Despite increases in progenitor frequency, total FL cellularity was significantly reduced in response to RH infection (Fig EV1M), and there was a virulence-dependent reduction in the overall number of CD45+ (hematopoietic) FL cells (Fig EV1N). Surprisingly, despite these dramatic changes in CD45+ cellularity, HSPC cellularity was maintained in Pru infection and increased in response to RH infection (Fig 1F). We observed decreased cellularity of LT-HSCs in response to RH infection (Fig 1G), and a virulence-dependent decrease in cellularity of ST-HSCs (Fig 1H), but no changes in specific Tom+ HSC and GFP+ drHSC populations (Fig 1I and J). Increased HSPC cellularity in response to RH infection was driven by expansion of MPP3 and MPP4 subsets, but not MPP2s (Fig 1K–M). Since overall HSPC numbers were maintained or expanded despite a decrease in CD45+ cellularity, we hypothesized that maternal infection promoted HSPC proliferation. A virulence-dependent increase in Ki67 expression in HSCs (CD150$^{hi}$) indicated that maternal infection with RH, but not Pru drove HSC proliferation (Fig 1N). MPPs (CD150− HSPCs) exhibited a trending increase in proliferation as determined by Ki67 expression (Fig 1O), supporting observations of an expanded MPP3 and MPP4 compartment (Fig 1E, L and M). We did not measure cell death; thus, it is possible that cell death contributed to a decrease in cellularity of LT- and ST-HSCs in response to RH infection. Nonetheless, severity of maternal infection appeared to drive immediate virulence-dependent changes in hematopoiesis by triggering HSC proliferation and expansion of downstream HSPCs in the fetal liver.

### In utero exposure to T. gondii modulates fetal HSC function

Next, we directly investigated the impact of maternal infection on HSC self-renewal and function by performing competitive transplantation assays following maternal infection with *T. gondii*. Acute infection and inflammation in adult hematopoiesis affects HSC self-renewal and biases HSC output toward the myeloid lineage (Pietras, 2017). To test the effect on fetal HSCs, we isolated and sorted Tom+ HSCs or GFP+ drHSCs from E15.5 FL following the maternal infection and competitively transplanted them with $5 \times 10^5$ adult whole bone marrow (WBM) cells into lethally irradiated adult recipients (Fig 2A). Recipient mice were monitored every 4 weeks for 16 weeks post-transplantation to determine long-term multilineage reconstitution (LTMR), or the ability of HSCs to reconstitute mature lineages in the peripheral blood, as well as

progenitors within the BM niche, in both primary and secondary recipients (Fig 2A).

Maternal infection with *T. gondii* influenced the long-term function of both Tom+ HSCs and GFP+ drHSCs upon transplantation (Fig 2B–J). The ratio of Tom+ recipients with sustained LTMR after primary transplantation was equivalent across all infection conditions (Fig 2B). However, maternal infection enhanced Tom+ HSC function by increasing myeloid output; peripheral blood (PB) granulocyte-macrophage (GM) chimerism was higher in response to both Pru and RH infections (Fig 2C), and platelet chimerism was also increased in response to Pru infection (Fig 2D). In contrast, there was a sharp reduction in the LTMR ratio among recipients of GFP+ drHSCs in response to RH when compared to saline control (Fig 2B), and maternal infection did not significantly affect GM or platelet chimerism in primary recipients of GFP+ drHSCs (Fig 2G and H). B- and T-cell chimerism in recipients of both Tom+ HSCs (Fig 2E and F) or GFP+ drHSCs (Fig 2I and J) was also unchanged in response to infection.

Eighteen weeks after primary transplantation, we investigated the long-term contribution of transplanted HSCs to progenitors in the bone marrow of primary recipients (Fig EV2). In parallel to PB output, prenatal exposure to Pru infection in recipients of Tom+ HSCs increased BM chimerism across most stem and progenitor cells, whereas HSPC chimerism in recipients of RH-exposed Tom+ HSCs was comparable to saline controls (Fig EV2A–F). We also observed expansion of bone marrow myeloid progenitors in Tom+ HSC recipients following Pru exposure, including increased chimerism of progenitors of granulocytes, macrophages, and megakaryocytes (Fig EV2G and H), but not erythroid progenitors (Fig EV2I). In contrast to Tom+ HSC recipients, infection did not increase BM chimerism in GFP+ drHSC recipients and was generally decreased in the RH-exposed recipients across all progenitor populations (Fig EV2A–M) indicating early exhaustion of GFP+ drHSCs in RH, but not Pru, recipients. Thus, greater virulence was clearly detrimental for the developmentally-restricted GFP+ drHSC, but enhanced output of the Tom+ HSC, a putative adult HSC precursor.

To further test HSC long-term self-renewal capability in response to infection, we performed secondary transplantation assays, in which whole bone marrow cells from primary recipients were transplanted into irradiated secondary recipients (Fig 2A). We transplanted twice as many WBM cells from GFP+ drHSC recipients, based on our previous report that GFP+ drHSCs exhibited less self-renewal potential in a secondary transplant setting (Beaudin et al, 2016). All Tom+ HSCs recipients retained robust LTMR in the peripheral blood upon secondary transplantation (Fig 2K) that was

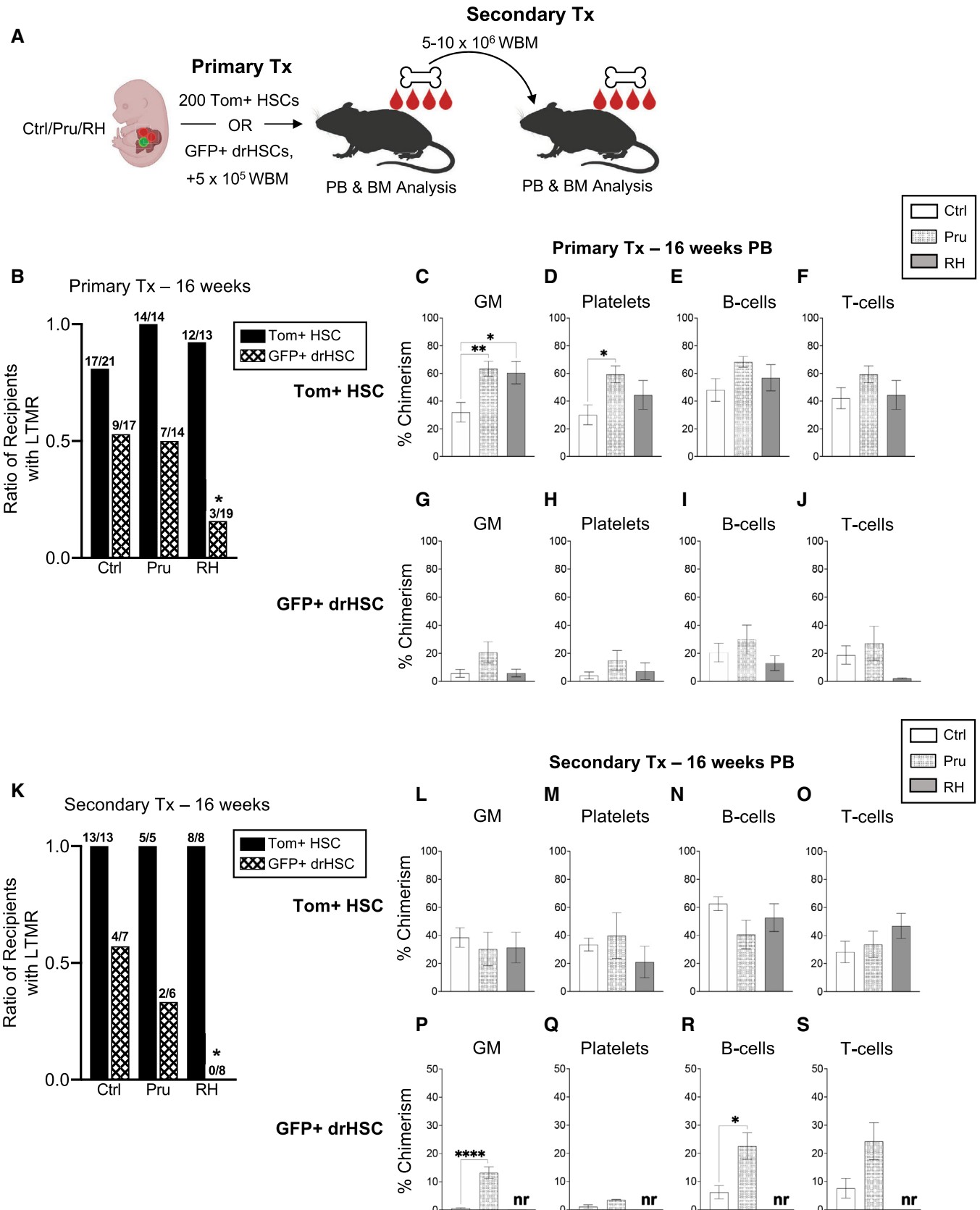

**Figure 2.**

**Figure 2. *In utero* exposure to *Toxoplasma gondii* modulates fetal HSC function.**

A   Schematic of transplantation experiments. After prenatal infection with Pru or RH strains, or saline control, injections as shown in Fig 1A, 200 E15.5 FL Tom+ HSCs or GFP+ drHSCs were isolated and transplanted into lethally irradiated WT recipients along with $5 \times 10^5$ WT whole bone marrow (WBM) cells. $5 \times 10^6$ WBM from fully reconstituted Tom+ HSC primary recipients or $1 \times 10^7$ WBM from GFP+ drHSC primary recipients were transplanted into lethally irradiated secondary recipients. Primary and secondary recipients were monitored for peripheral blood output of mature blood cell populations and bone marrow chimerism.

B   Ratio of primary recipients from (A) with long-term multi-lineage reconstitution (LTMR: ≥ 1% peripheral blood chimerism within each mature lineage) in all four lineages (granulocyte/macrophages [GM], platelets, B- and T-cells), *n* is shown as the numerator. Statistical significance determined by Fisher's exact test (two-tailed). *$P \leq 0.05$.

C–F   Peripheral blood (PB) chimerism of primary Tom+ HSC transplant recipient mice with LTMR (B) in (C) granulocytes/macrophages (GM), (D) platelets, (E) B-cells, and (F) T-cells at Week 16 post-transplantation.

G–J   Peripheral blood (PB) chimerism of primary GFP+ drHSC transplant recipient mice with LTMR (B) in (G) granulocytes/macrophages (GM), (H) platelets, (I) B-cells, and (J) T-cells at Week 16 post-transplantation.

K   Ratio of secondary recipients from (A) with long-term multi-lineage reconstitution (LTMR; ≥ 0.1% PB chimerism) in all four lineages (GM, platelets, B- and T-cells), *n* is shown as the numerator. Statistical significance determined by Fisher's exact test (two-tailed). *$P \leq 0.05$.

L–O   Peripheral blood (PB) chimerism of secondary recipients of Tom+ HSC with LTMR (K) in (L) granulocytes/macrophages (GM), (M) platelets, (N) B-cells, and (O) T-cells at Week 16 post-transplantation.

P–S   Peripheral blood (PB) chimerism of secondary recipients of GFP+ drHSCs with LTMR (K) in (P) granulocytes/macrophages (GM), (Q) Platelets, (R) B-cells, and (S) T-cells at Week 16 post-transplantation. No reconstitution (nr) was present in recipients of GFP+ drHSCs in RH.

Data information: Graphs (C–J) and (L–S) are plotted as the mean ± SD. One-way ANOVA with Tukey's test. *$P \leq 0.05$; **$P \leq 0.01$; ****$P \leq 0.0001$.
Source data are available online for this figure.

equivalent across all PB mature cell subsets regardless of virulence (Fig 2L–O). Robust PB chimerism was mirrored by comparable chimerism across most BM stem, progenitor, and mature cell compartments (Fig EV2N–Z), with significantly higher chimerism only in common lymphoid progenitors (CLPs) (Fig EV2X). In contrast, GFP+ drHSC LTMR was abolished in response to the more virulent infection, with 0/8 GFP+ drHSC recipients exhibiting LTMR in the RH condition (Fig 2K). Surprisingly, although only 2/6 recipients were reconstituted upon secondary transplantation with Pru-exposed GFP+ drHSCs, chimerism among reconstituted recipients was significantly higher than controls for GMs (Fig 2P), but not platelets (Fig 2Q). Similarly, recipient chimerism was also significantly higher for B-cell (Fig 2R) lineages, with no differences in T-cell lineages (Fig 2S). Analysis of BM chimerism in secondary recipients confirmed that only GFP+ drHSC recipients in the Pru condition exhibited any progenitor chimerism, whereas recipients of RH-exposed GFP+ drHSCs had no detectable BM chimerism (Fig EV2N–Z), consistent with the absence of LTMR ("nr" or no reconstitution) (Fig 2K and P–S). Tom+ HSC recipients therefore maintained LTMR without loss of self-renewal potential in response to maternal toxoplasma infection, regardless of virulence, while GFP+ drHSCs succumbed to exhaustion in a virulence-dependent manner.

## Maternal *T. gondii* infection increases inflammatory cytokines in the fetus

To gain further insight into the mechanisms underlying the effects of maternal infection on fetal hematopoiesis, we characterized inflammation in the amniotic fluid and fetal liver for levels of inflammatory cytokines at E15.5 (Fig 3). In the fetal amniotic fluid (Fig 3A–J), infection with either Pru or RH induced high levels of IFNγ, IFNβ, IL-1β, and IL-10 as compared to saline controls (Fig 3A–D). Upregulation of other cytokines in fetal amniotic fluid was virulence-dependent, including significant increases in IL-1α (Fig 3E), TNF-α (Fig 3F), and IL-6 (Fig 3G). only after maternal infection with the more virulent strain, RH. Two cytokines, IL-27 (Fig 3H) and GM-CSF (Fig 3I), were decreased in amniotic fluid in response to infection. Within the fetal liver supernatant (Fig 3K–T),

while there were measurable levels of IFNγ (Fig 3K), we observed a significant virulence-dependent upregulation of IL-1α in the fetal liver (Fig 3O) and a significant increase in IL-1β (Fig 3M) for both *T. gondii* strains. In contrast to amniotic fluid, IFN-β (Fig 3L), IL-10 (Fig 3N), IL-6 (Fig 3Q), and IL-27 (Fig 3R) levels were undetectable in response to infection, and TNF-α (Fig 3P) levels were decreased albeit barely above the limit of detection. GM-CSF was not detectable in the fetal liver (Fig 3S). Comparison of maternal serum levels (Fig 3U–E′) in response to infection revealed increased levels of IFNγ (Fig 3U and E′), IL-10 (Fig 3X), TNFα (Fig 3Z), IL-6 (Fig 3A′), and IL-27 (Fig 3B′). Surprisingly, IFNγ, IL-10, and IL-27 levels were only significantly elevated in response to the less virulent Pru infection at this timepoint. IL-6 was only increased in maternal serum in response to RH infection (Fig 3A′). As compared to the fetal response, neither IFNβ (Fig 3V), IL-1β (Fig 3W), IL-1α (Fig 3Y), nor GM-CSF (Fig 3C′) showed a significant response to infection in maternal serum, suggesting that these cytokines are unique to the fetal hematopoietic response to maternal infection. Thus, while IFNγ is a significant component of cytokine activity in the fetus, *T. gondii* infection also causes the upregulation of other cytokines within the fetal environment, which may affect fetal hematopoiesis.

## Prenatal exposure to IFNγ activates fetal hematopoiesis

As maternal infection elicited a broad and diverse proinflammatory cytokine response in the fetus (Fig 3), we sought to disentangle the immediate fetal hematopoietic response to maternal infection by focusing on the impact of a single cytokine, IFNγ. The maternal response to *T. gondii* infection is marked by a robust increase in IFNγ in the serum in response to less virulent Pru infection (Fig 3U and E′) and IFNγ is also present in the fetal microenvironment during maternal infection (Fig 3A and K). IFNγ is also a critical regulator of the fetal response to *T. gondii* during pregnancy (Shiono et al, 2007), and independently regulates HSC emergence during development (Li et al, 2014). We hypothesized that fetal HSCs may be directly responsive to IFNγ during infection, as observed for adult HSCs (Baldridge et al, 2010; Morales-Mantilla & King, 2018). To test the immediate response of fetal HSPCs to IFNγ, we injected

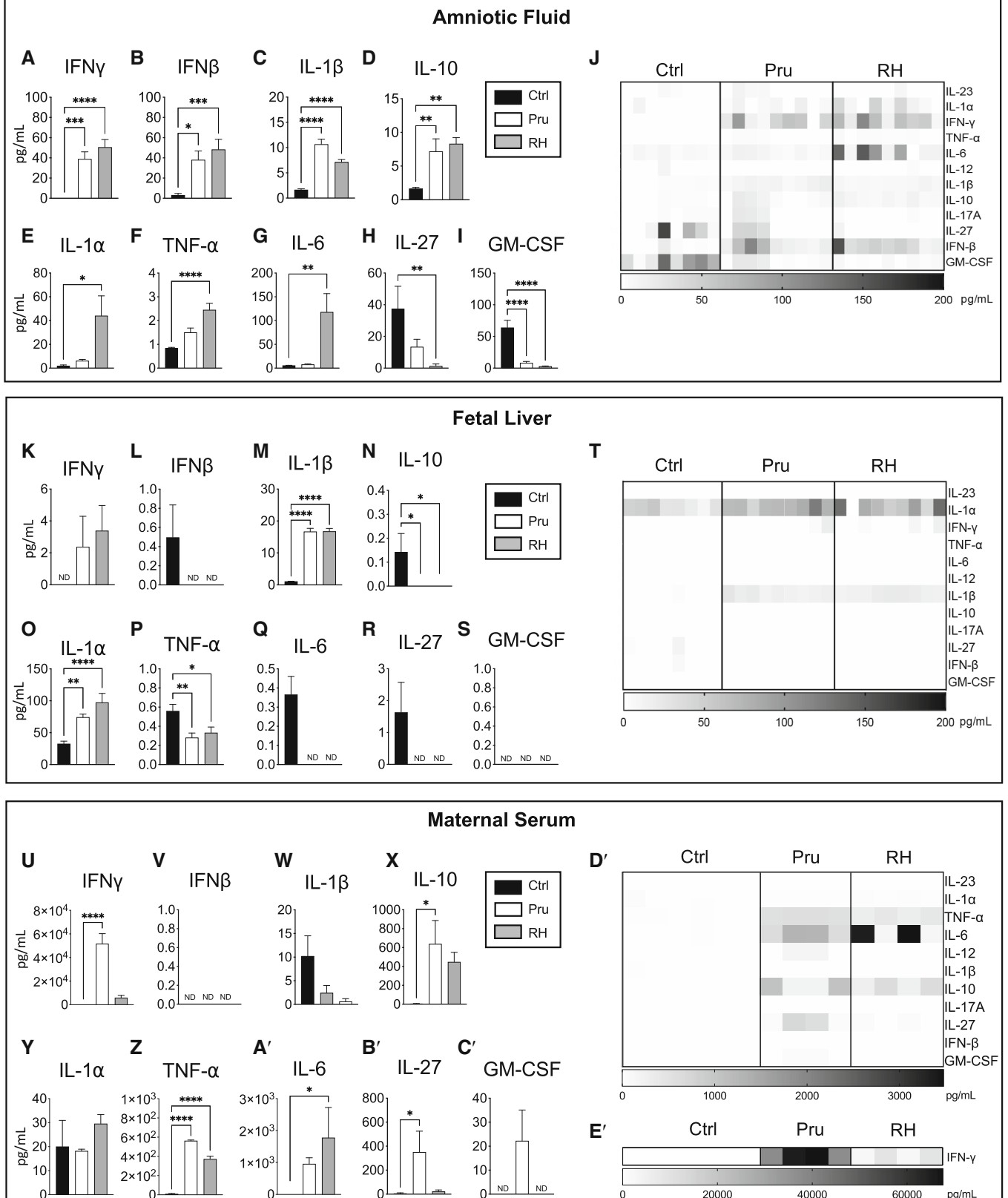

**Figure 3.**

**Figure 3.  *Toxoplasma gondii* virulence from maternal infection modulates inflammation in the fetal environment.**

A–I     Measurement of (A) IFNγ, (B) IFNβ, (C) IL-1β, (D) IL-10, (E) IL-1α, (F) TNFα, (G) IL-6, (H) IL-27, and (I) GM-CSF cytokines in fetal amniotic fluid at E15.5 following pre-
        natal infection. *n* = 8–9 fetuses from at least 3 litters/condition.
J       Heatmap of inflammatory cytokines in fetal amniotic fluid at E15.5. Each column represents the results from an individual fetus at E15.5 following prenatal
        infection.
K–S     Measurement of (K) IFNγ, (L) IFNβ, (M) IL-1β, (N) IL-10, (O) IL-1α, (P) TNFα, (Q) IL-6 (R) IL-27, and (S) GM-CSF cytokines in fetal liver supernatant at E15.5 following
        prenatal infection. *n* = 8–9 fetuses from at least 3 litters/condition.
T       Heatmap of inflammatory cytokines in the fetal liver at E15.5. Each column represents the results from an individual fetus at E15.5 following prenatal infection.
U–C′    Measurement of (U) IFNγ, (V) IFNβ, (W) IL-1β, (X) IL-10, (Y) IL-1α, (Z) TNFα, (A′) IL-6 (B′) IL-27, and (C′) GM-CSF cytokines in maternal serum at E15.5 following pre-
        natal infection; *n* = 3 mice/condition.
D′      Heatmap of inflammatory cytokines in maternal serum at E15.5. Each column represents the results from individual dams at E15.5 following prenatal infection.
E′      Heatmap of IFNγ in maternal serum at E15.5. Each column represents the results from individual dams at E15.5 following prenatal infection.

Data information: For all analysis above bars represent mean ± SD. One-way ANOVA with Tukey's test. *$P \leq 0.05$; **$P \leq 0.01$; ***$P \leq 0.001$; ****$P \leq 0.0001$.
Source data are available online for this figure.

20 μg of recombinant IFNγ into pregnant (E14.5) WT dams mated to FlkSwitch mice and examined the fetal hematopoietic stem and progenitor response 1 day later at E15.5. A single injection of maternal IFNγ did not have overt physical effects on fetal development, as crown–rump length was unaffected in E15.5 fetuses. Despite large differences in maternal serum and amniotic fluid levels of IFNγ induced by *T. gondii* infection (Fig 3A and U) versus direction injection of IFNγ (Fig EV4B and E), both manipulations resulted in similar levels of IFNγ in the fetal liver supernatant (Figs 3K and EV4D). At E15.5, HSPC numbers increased significantly in response to maternal IFNγ (Fig 4A). Expansion was observed across all HPSC subsets, including LT-HSC (Fig 4B), fractionated Tom+ HSCs and GFP+ drHSCs (Fig 4C and D), ST-HSCs (Fig 4E) and all downstream MPP subsets (Fig 4F–H). These data demonstrate that fetal HSPCs are sensitive to even moderate levels of IFNγ, and may drive expansion in response to virulent *T. gondii* infection.

### Prenatal exposure to IFNγ enhances fetal hematopoietic stem cell function and long-term multi-lineage reconstitution

As prenatal IFNγ injection induced a response across all HSPC subsets, we sought to isolate the effects of IFNγ on HSC function. We isolated and transplanted 200 Tom+ HSCs or GFP+ drHSCs with $5 \times 10^5$ whole bone marrow cells into lethally irradiated primary recipients and assessed peripheral blood and bone marrow progenitor reconstitution in primary and secondary recipients. Prenatal administration of IFNγ did not affect the ratio of recipients with long-term multilineage reconstitution for either Tom+ HSCs or GFP+ drHSCs in primary recipients (Fig 5A). Importantly, IFNγ exposure increased peripheral blood chimerism in primary recipients of both Tom+ HSC and GFP+ drHSC for GM, platelets, and B-cell lineages, but not T-cells (Fig 5B–I). Compared with saline-treated controls, primary recipients of IFNγ-exposed GFP+ drHSCs exhibited significantly increased chimerism across all HSPC populations in the bone marrow at 18 weeks (Fig EV3A–F), as well as significantly increased chimerism of megakaryocyte progenitors (MkPs; Fig EV3H) and mature myeloid cells (GMs; Fig EV3J). Primary recipients of Tom+ HSCs also exhibited a more modestly increased profile of BM chimerism in response to IFNγ exposure, with significant increases in LT-HSCs (Fig EV3B), MPP3s (Fig EV3E), and MkPs (Fig EV3H). Chimerism of BM lymphoid cells, including CLP (Fig EV3K) and B- and T-cells (Fig EV3L and M), as well as GMP and EP (Fig EV3G and I) were unaffected by IFNγ exposure across all recipients. The effect of

direct injection of IFNγ therefore mimicked aspects of the intermediate Pru infection.

In secondary recipients, IFNγ treatment did not substantially affect the frequency of LTMR in either Tom+ HSC or GFP+ drHSCs (Fig 5J). While PB chimerism for GM, platelets, and B-cells was unchanged (Fig 5K–M), T-cell chimerism increased from Tom+ HSCs (Fig 5N). PB chimerism was also unaffected by IFNγ treatment in secondary recipients of GFP+ drHSCs (Fig 5O–R), and was overall very low. Analysis of BM chimerism in secondary recipients of Tom+ HSCs revealed that, compared to saline controls, chimerism was generally equivalent across all conditions (Fig EV3N–W and Y) except for an increase in CLPs (Fig EV3X) and T-cells (Fig EV3Z), that mirrored increased T-cell output in the PB (Fig 5N). In secondary recipients of GFP+ drHSCs, BM chimerism was increased in ST-HSCs (Fig EV3P), MPP2s (Fig EV3Q), and MPP3s (Fig EV3R) in response to IFNγ exposure. Thus, prenatal exposure to IFNγ alone increased self-renewal capacity and output of both Tom+ HSCs and transient GFP+ drHSCs upon primary transplantation, similar to our observations with the Pru infection of intermediate virulence.

### Fetal HSCs respond directly to maternal IFNγ through the IFNγ receptor

The differences in the response of HSCs and progenitors to IFNγ alone compared to IFNγ in the context of a complex infection led us to investigate the degree to which maternal or fetal IFNγ was responsible for the hematopoietic changes induced by *T. gondii* infection. We first investigated whether maternal sources of IFNγ crossed the placenta into the fetus. Pregnant IFNγ knock-out (IFNγ−/−) dams were injected intraperitoneally at E14.5 with 20 μg of recombinant IFNγ. One day after injection, cytokine analysis of both fetal amniotic fluid and fetal liver revealed the presence of IFNγ in IFNγKO fetuses (Fig 6A), demonstrating for the first time that maternal IFNγ crosses the maternal–fetal barrier. LT-HSCs in the fetal liver express the IFNγ receptor (IFNγR) (Baldridge *et al*, 2010) and can therefore directly sense IFNγ. To investigate the fetal HSC response via the IFNγ receptor, IFNγR +/− dams were time mated with IFNγR −/− males (Fig 6B). Pregnant dams were injected with 20 μg of IFNγ at E14.5, and fetal liver HSPCs were quantified at E15.5. Surprisingly, prenatal IFNγ exposure induced the same expansion of total HSPCs in both IFNγR +/− and −/− fetuses (Fig 6C), despite the absence of the IFNγR in −/− fetuses. Interestingly, IFNγR deletion on the fetal side resulted in

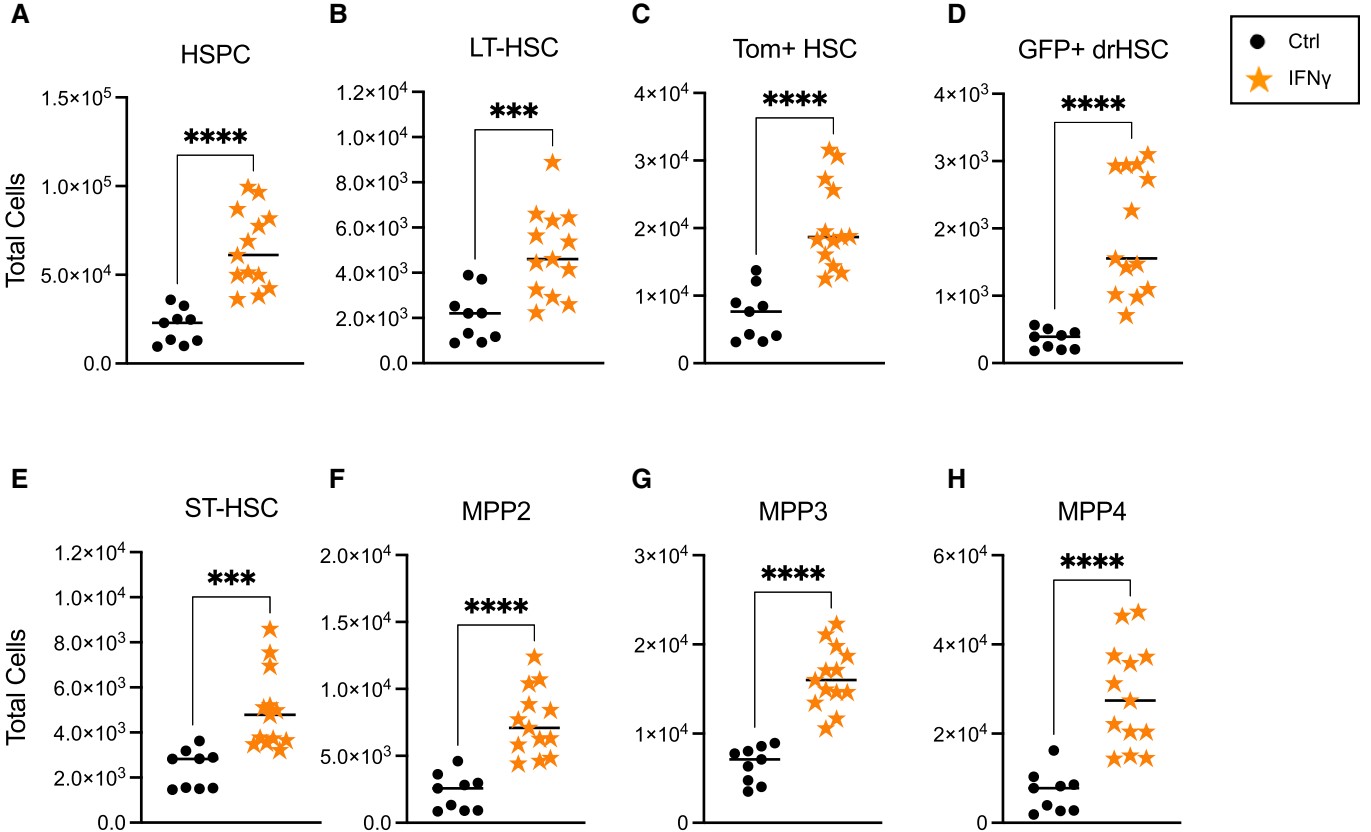

**Figure 4. Prenatal exposure to IFNγ activates fetal hematopoiesis.**

A–H  Total cellularity of (A) HSPCs, (B) LT-HSCs, (C) Tom+ HSCs, (D) GFP+ drHSCs, (E) ST-HSCs, (F) MPP2, (G) MPP3, and (H) MPP4 at E15.5 following maternal IFNγ injection. For all analysis above, bars represent mean ± SEM. *n* = 9–13 fetuses from 3–4 litters/condition. Statistical significance was determined by unpaired Student's *t*-test. ***$P \leq 0.001$; ****$P \leq 0.0001$.

Source data are available online for this figure.

accumulation of IFNγ in amniotic fluid and fetal liver even in the absence of exogenous IFNγ administration, suggesting accumulation of cytokine due to lack of receptor signaling (Fig EV4A–D). Moreover, detection of IFNγ on the fetal side in the absence of IFNγR at the maternal-fetal interface indicated significant passive transport of IFNγ across the placenta that is independent of receptor-mediated transcytosis. When further examined across HSPC populations, deletion of the fetal IFNγR abrogated the expansion of both LT-HSCs (Fig 6D) and ST-HSCs (Fig 6E) in response to maternal IFNγ. Surprisingly, however, maternal IFNγ treatment still expanded MPP populations even in IFNγR −/− fetuses (Fig 6F–H). These data suggest that while IFNγ acts directly upon fetal HSCs, downstream activation and expansion of other HSPC populations is dissociated from HSC activation and may be driven by other cytokines produced downstream of maternal IFNγR signaling.

To further analyze the fetal HSC response to IFNγ, we performed the opposite cross, wherein IFNγR −/− dams were time mated with IFNγR +/− males (Fig 6I). In the absence of maternal IFNγ-signaling, fetal liver HSPCs remained expanded in response to IFNγ injection in fetuses with an intact copy of receptor (IFNγR +/−) (Fig 6J). Deletion of the maternal IFNγR eliminated the cellular response of LT-HSCs and ST-HSCs in both IFNγR +/− and −/−

fetuses (Fig 6K and L). Despite the inability of the dam to mount an IFNγ-signaling response (Fig EV4E), all MPP subsets were expanded in response to maternal injection with IFNγ (Figs 6M–O and EV4F). Collectively, the data would suggest that IFNγ-signaling is necessary in both the fetus and dam to elicit a response in LT-HSCs and ST-HSCs. These data also imply that the immediate MPP response is distinct from the fetal HSC response, which requires both direct IFNγ-signaling and a maternal derived IFNγ-induced factor. Together, these data provide direct evidence that the fetal hematopoietic response therefore coordinates signals across the maternal–fetal interface and reflects the dynamic and diverse responses of distinct HSPCs.

## Discussion

Our investigation reveals the multifaceted response of fetal hematopoietic stem and progenitor cells (HSPCs) to a complex and acute prenatal infection. *T. gondii* is a ubiquitous pathogen and infection during pregnancy has severe implications for fetal health and development. Inflammation from prenatal infections such as *T. gondii* drive alterations to fetal immune development and shape the

postnatal immune response in offspring (López et al, 2022), but the cellular mechanisms underlying those changes are poorly understood. Here, we demonstrate that fetal HSPCs are directly affected by prenatal infection, and that both maternal and fetal responses to prenatal infection affect proliferation, self-renewal, and lineage output of fetal HSCs. Our results suggest that the response of fetal HSCs to infection is a novel mechanism by which prenatal infection drives changes in postnatal hematopoietic output and function in offspring.

*Toxoplasma gondii* elicits a typical Type II IFNγ-mediated host immune response to resolve infection and clear parasites (Sturge & Yarovinsky, 2014). During gestation, the typified IFNγ-mediated response has been demonstrated to limit vertical transmission at the expense of fetal growth and survival (Abou-Bacar et al, 2004; Shiono et al, 2007; Pappas et al, 2009; Senegas et al, 2009). This well-characterized immune interplay, along with *T. gondii*'s established role in congenital infections, makes *T. gondii* an ideal model of prenatal infection. Both *T. gondii* strains, Pru and RH, caused

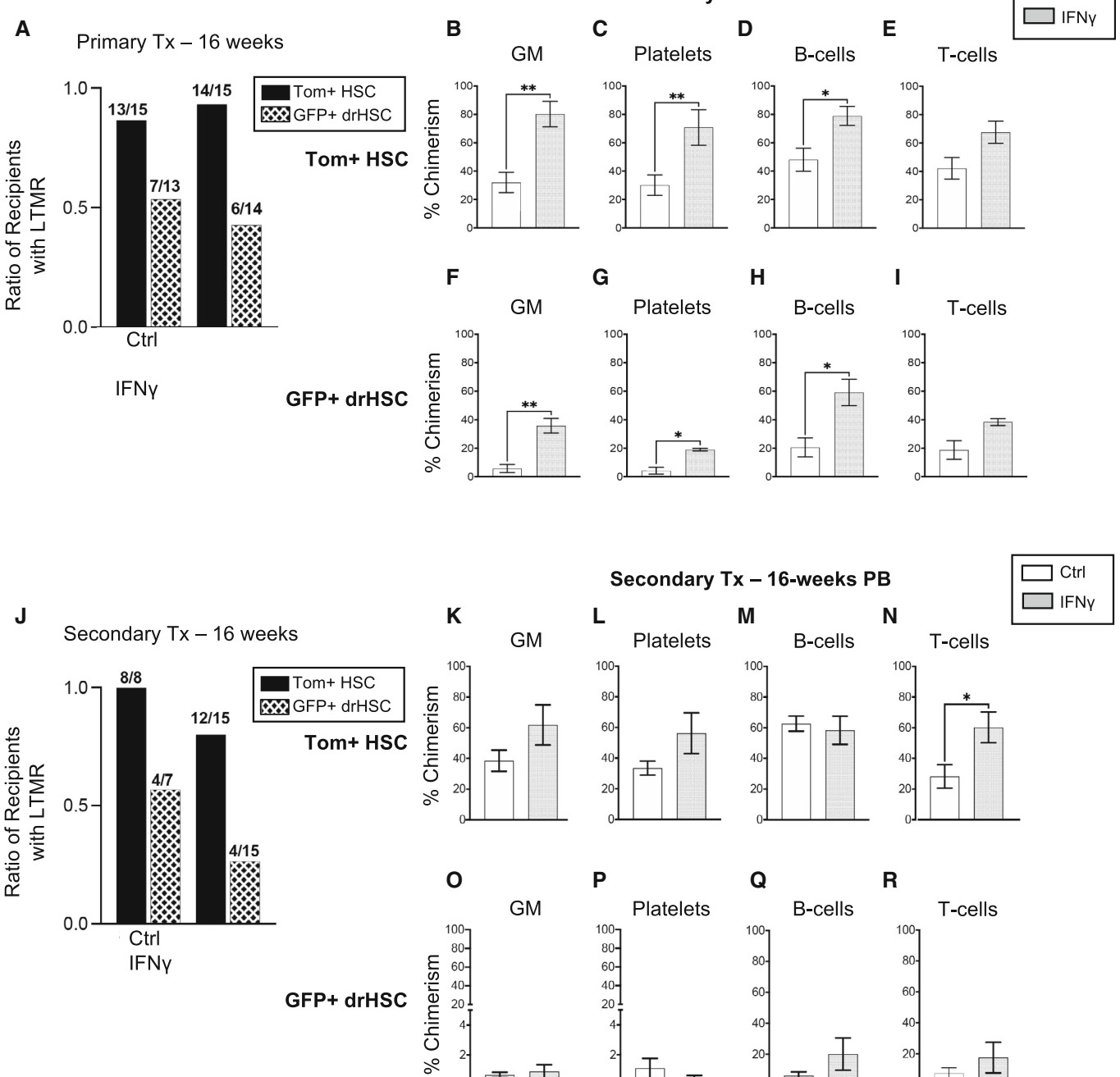

**Figure 5.**

**Figure 5.** **Prenatal exposure to IFNγ enhances fetal HSC function.**

A   Ratio of recipients with long-term multilineage reconstitution (LTMR: ≥ 1% PB chimerism) in all four lineages (GM, platelets, B- and T-cells). *N* of mice is shown as the numerator.

B–E   Peripheral blood (PB) chimerism of (B) granulocyte/macrophages (GM), (C) platelets, (D) B-cells, and (E) T-cells in Tom+ HSC transplant recipients in (A) at 16 weeks of post-transplantation.

F–I   Peripheral blood (PB) chimerism of (F) granulocyte/macrophages (GM), (G) platelets, (H) B-cells, and (I) T-cells in GFP+ drHSC transplant recipients of mice in (A) at 16 weeks of post-transplantation.

J   Ratio of secondary transplant recipients with long-term multi-lineage reconstitution (LTMR ≥ 0.1% PB chimerism) in all four lineages (GM, platelets, B- and T-cells). *N* of mice is shown as the numerator.

K–N   Peripheral blood (PB) chimerism of (K) granulocyte/macrophages (GM), (L) platelets, (M) B-cells, and (N) T-cells in secondary Tom+ HSC transplant recipients in (J) at 16 weeks of post-transplantation.

O–R   Peripheral blood (PB) chimerism of (Q) granulocyte/macrophages (GM), (R) Platelets, (S) B-cells, and (T) T-cells in secondary transplant recipients of GFP+ drHSCs in (J) at 16 weeks of post-transplantation. For all % chimerism experiments, mice not demonstrating LTMR were not included in analysis.

Data information: Bars represent mean ± SD. Statistical significance was determined by unpaired Student's *t*-test. *$P ≤ 0.05$; **$P ≤ 0.01$.

Source data are available online for this figure.

significant fetal growth restriction, but the RH strain induced a broader inflammatory response in the fetus, as indicated by high levels of IL-6 and IL-1α in the amniotic fluid. We did not directly assess vertical transmission in this study; however, given the degree of virulence, it would be expected that the RH strain would be vertically transmitted more frequently than Pru infections. Adoptive transfer experiments confirmed that HSPCs themselves were not directly infected, as transplantation of cells infected with even one RH parasite would have resulted in immediate death (Pfefferkorn & Pfefferkorn, 1976; Howe *et al*, 1996). In this context and given the degree of growth restriction and inflammation induced by congenital RH infection, it is not surprising that we observed a stronger response from fetal HSPCs to congenital RH infection *in situ*.

Using a poly(I:C)-induced model of maternal immune activation (MIA), we have recently demonstrated that Type I interferon-mediated prenatal inflammation can drive lasting changes in postnatal immune function by specifically activating lymphoid-biased drHSCs (López *et al*, 2022). In this simplified model of prenatal inflammation, we further demonstrated that the GFP+ drHSC was highly responsive to Type I interferons, causing its inappropriate expansion and persistence into the postnatal period. Our present investigation revealed that Type II IFNγ-mediated inflammation, both in the context of an actual congenital infection and when isolated independently, did not differentiate responses between the Tom+ HSC and GFP+ drHSC *in situ*. Previous work has shown that the effects of an infection on adult hematopoiesis can differ greatly from the impact of a single cytokine. For example, adult BM HSCs respond very differently to direct infection with cytomegalovirus and vesicular stomatitis virus, viruses that induce a typical Type-1 interferon response, as compared to direct injection with interferon-inducing reagents, such as poly(I:C) (Hirche *et al*, 2017). Perhaps not surprisingly, despite induction of comparable IFNα levels, viral infections were found to evoke complex inflammatory responses that activated HSCs in a manner independent of IFNAR signaling. To resolve the extent to which the fetal hematopoietic response to maternal *T. gondii* infection was driven by IFNγ, the primary cytokine mediating immune clearance in *T. gondii* infection (Suzuki *et al*, 1988), we compared fetal hematopoiesis after prenatal exposure to both IFNγ and infection with two strains of *T. gondii* of increasing virulence. Direct injection of IFNγ induced effects on fetal HSCs upon transplantation that were more similar to those observed in response to the less virulent Pru infection. Hence, it plausible that

with a less virulent infection, maternally-derived IFNγ may be a more prominent driver of the HSC response, whereas with more virulent infections other cytokines, such as IL-6 and IL-1α, may further impact HSC function (Mirantes *et al*, 2014). Indeed, IFNγ was highly upregulated in maternal serum in response to Pru infection. Although IL-1α and IL-1β were more significantly increased in the fetal liver as compared to IFNγ in response to infection, our previous transcriptional analysis of fetal HSPCs suggested that fetal HSPCs were not highly responsive to IL-1 signaling (López *et al*, 2022).

As opposed to the effect of prenatal infection and direct maternal injection of IFNγ on hematopoiesis *in situ*, adoptive transfer assays revealed disparate responses of two fetal HSC populations differentiated by the FlkSwitch model: the Tom+ HSC as a more "conventional" putative adult HSC precursor, and the GFP+ Flk2-marked HSC as a transient, lymphoid-biased HSC that functions to give rise to innate-like lymphocytes during fetal development (Beaudin *et al*, 2016). Whereas the distinct effects on each population may be more difficult to parse out among varied responses of HSPCs *in vivo*, adoptive transfer assays allow dissection of the discrete outcomes across pecific cell types. Tom+ HSCs demonstrated both increased myeloid output upon primary transplantation and sustained output in a secondary transplant setting, even in response to a highly virulent infection. While a myeloid-biased response is akin to observations in adult HSCs in response to infection (Baldridge *et al*, 2010; Matatall *et al*, 2014; Haas *et al*, 2015) sustained output following acute inflammation may be unique to fetal HSCs. In contrast, prenatal infection had a virulence-dependent deleterious effect on the function of the GFP+ drHSC upon transplantation. The less virulent Pru infection increased myeloid output resulting in a loss of lymphoid bias, and significantly increased GM and B-cell output in recipients that engrafted upon secondary transplantation. The more virulent RH infection impaired LTMR in primary transplantation and resulted in complete exhaustion in secondary transplantation. The disparate response of two fetal HSC populations may reflect the differential sensitivity to distinct inflammatory mediators induced by congenital infection. The response of the Tom+ HSC is more consistent with that of the adult, whereas the response of the GFP+ drHSC may be more typical of transient progenitors that may rapidly differentiate to support the immunological needs against infection. Nonetheless, transplantation outcomes may also reflect their

Diego A López et al

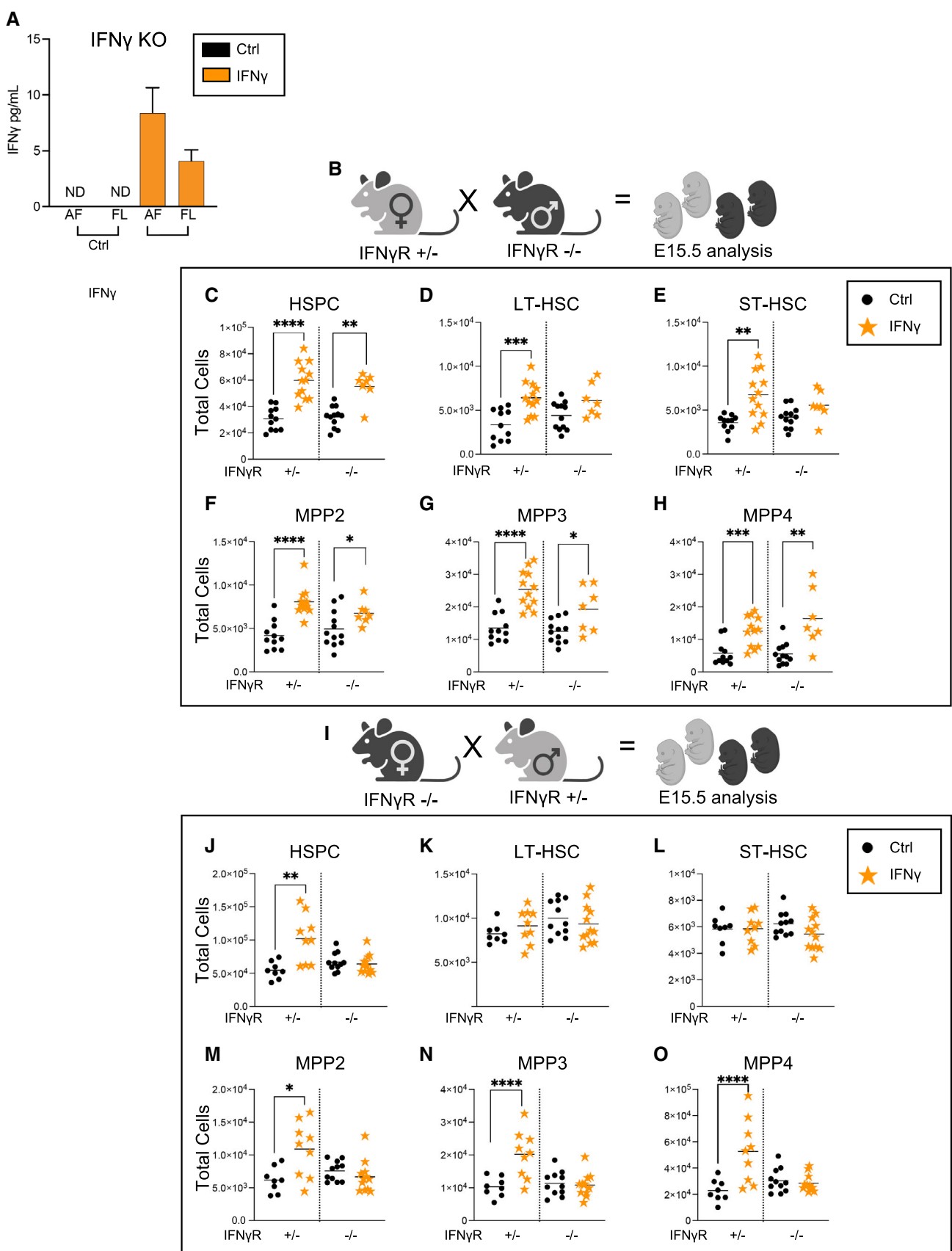

**Figure 6.**

**Figure 6.  Fetal HSCs respond directly to maternal IFNγ through the IFNγ receptor.**

A    IFNγ cytokine measured in the amniotic fluid (AF) and fetal liver supernatant (FL) of IFNγKO (IFNγ−/−) mice 1 day following injection of cytokine on E14.5. ND = not detectable. Data represent mean of three fetuses/condition and error bars represent ± SD.

B    Schematic of timed matings. IFNγR +/− dams were time mated to IFNγR −/− males and injected with saline or 20 μg IFNγ at E14.5. Fetal liver HSPCs were analyzed at E15.5.

C–H  Total cellularity of (C) HSPCs, (D) LT-HSCs, (E) ST-HSCs, (F) MPP2, (G) MPP3, and (H) MPP4 in saline or IFNγ exposed littermates of each genotype (IFNγ +/− or −/−), as derived from the cross as shown in (B) (IFNγR +/− dam). *n* = 19–23 fetuses from 3 litters/condition.

I    Schematic of timed matings. IFNγR −/− dams were time mated to IFNγR +/− males and injected with saline or 20 μg IFNγ at E14.5. Fetal liver HSPCs were analyzed at E15.5.

J–O  Total cellularity of (J) HSPCs, (K) LT-HSCs, (L) ST-HSCs, (M) MPP2, (N) MPP3, and (O) MPP4 in saline or IFNγ exposed littermates of each genotype (IFNγ +/− or −/−), as shown in (I) (IFNγR −/− dam). *n* = 19–21 fetuses from 3 litters/condition.

Data information: For all analysis bars represent mean. Statistical significance was determined by unpaired Student's *t*-test. *$P \leq 0.05$; **$P \leq 0.01$; ***$P \leq 0.001$; ****$P \leq 0.0001$.

Source data are available online for this figure.

unique capacity when placed within the bone marrow microenvironment.

To begin to dissect the requirement for IFNγ responsiveness across the maternal–fetal interface, we examined the effect of direct injection of IFNγ in crosses of IFNγR −/− mice. We first confirmed that maternal IFNγ can directly cross the placenta, and also demonstrated that while the fetal IFNγR is required for the HSC response, it is not required for the immediate expansion of more differentiated MPPs. These data have two important implications. First, the persistent response of MPPs in the absence of the fetal IFNγR and the elimination of a direct response of upstream HSCs indicates that MPPs respond to distinct inflammatory cues compared to HSCs, including downstream cytokines induced by maternal IFNγR signaling, such as IL-6 (Biondillo *et al*, 1994). This finding suggests a highly orchestrated fetal hematopoietic response to inflammation. Second, the expansion of fetal MPPs in response to exogenous IFNγ, even when IFNγR was deleted on the maternal side, indicates that the fetus can mount its own IFNγ-dependent cytokine response capable of activating fetal hematopoietic progenitors independent of the maternal response. Maternal IFNγ signaling still regulates the fetal response, as maternal deletion of IFNγR impaired fetal HSC expansion, even in fetuses with an intact receptor. The identity of additional IFNγ-induced factors has yet to be experimentally determined. Another interesting observation was that lack of IFNγ receptor on either the fetal or maternal side caused accumulation of IFNγ in fetal tissues, likely due to lack of uptake by the receptor (Marchetti *et al*, 2006). These interactions illustrate the complex interface between the fetal and maternal response that is mediated not only by the broad and diverse cytokine response of the mother, but also the discrete and independent cytokine response initiated by the fetus.

Our study is the first of its kind to explore the impact of maternal infection on fetal hematopoiesis. By comparing the effects of a multifaceted infection to the effects of a single cytokine injection on fetal HSPCs, our work begins to narrow down the inflammatory mediators of the hematopoietic response to infection. We also leveraged genetic models to determine whether maternal signals, such as IFNγ, cross the fetal–maternal interface and interact directly with fetal hematopoietic cells to drive the hematopoietic response. Our model of maternal infection limited our ability to investigate the effects on postnatal hematopoiesis, as the interferon response induced upon acute infection in mice generally causes fetal demise (Yockey & Iwasaki, 2018). Instead, we used adoptive transfer as a proxy to investigate the long-term functional impact to HSCs. Still,

we demonstrate that maternally derived IFNγ acts directly on fetal HSCs to shape their long-term hematopoietic output without affecting their self-renewal potential. Our data therefore provide a novel mechanism whereby congenital infection might shape postnatal immune responses in offspring by affecting the function and output of developing hematopoietic stem cells.

# Materials and Methods

## Mouse models and husbandry

All mice were maintained in the University of California, Merced and University of Utah vivariums according to Institutional Animal Care and Use Committee (IACUC)-approved protocols.

Eight to twelve-week-old female C57BL/6 (RRID: IMSR_JAX:000664) mice were mated to male FlkSwitch mice (Boyer *et al*, 2011; Beaudin *et al*, 2016). At gestation day 14.5, pregnant dams were injected intraperitoneally with 20 μg recombinant IFNγ (Peprotech). IFNγ knock-out mice (JAX 002287) were used to assess IFNγ transfer across the fetal-maternal interface. To investigate the fetal HSC response via the IFNγ receptor, IFNγ receptor knock-out (IFNγR KO) mice (JAX 003288) were bred with wild-type C57BL/6J (WT) mice to generate mice with one copy of IFNγR (+/− mice). All saline controls were collected in parallel and published previously in (López *et al*, 2022). Pregnant dams were euthanized, and fetuses were dissected from the uterine horn. Fetal liver GFP+ (Flkswitch) expression was confirmed by microscopy.

## Parasite strains and peritoneal injections

*Toxoplasma gondii* strains, type I RH *Δku80 Δhxgprt* (Huynh & Carruthers, 2009) (LD$_{100}$ < 10) and type II Pru *Δhxgprt* (LD$_{50}$ = $10^2$–$10^4$) (Saeij *et al*, 2005; Kim *et al*, 2007), were grown in human foreskin fibroblasts (HFFs) and tachyzoites were prepared as described previously (Kongsomboonvech *et al*, 2020) and administered into pregnant dams via intraperitoneal injection of $2 \times 10^4$ tachyzoites in a volume of 100 μl in 1× PBS.

## Cell isolation and identification by flow cytometry

Fetal livers were dissected and pipetted gently in staining media to form a single-cell suspension. Cell populations were analyzed using a four-laser FACS Aria III (BD Biosciences) and five-laser Aurora

(Cytek). Cells were sorted on the FACS Aria II (BD Biosciences) or III. All flow cytometric analysis was done using FlowJo™. Hematopoietic and mature blood cell populations were identified as follows: Lineage dump or "Lin" for all fetal liver populations (CD3, CD4, CD5, CD8, CD19, Ter-119, Nk1.1, Gr-1, F4/80), LT-HSCs (Lin−, CD45+, cKit+, Sca1+, Flk2−, CD48−, CD150+), ST-HSCs (Lin−, CD45+, cKit+, Sca1+, Flk2−, CD48−, CD150−), Tom+ HSCs, (Lin−, CD45+, cKit+, Sca1+, CD150+, Tom+), GFP+ drHSCs (Lin−, CD45+, cKit+, Sca1+, CD150+, GFP+), MPP2 (Lin−, CD45+, cKit+, Sca1+, Flk2−, CD48+, CD150+), MPP3 (Lin−, CD45+, cKit+, Sca1+, Flk2−, CD48+ CD150−), MPP4 (Lin−, CD45+, CKit+, Sca1+, Flk2+), Granulocyte-Macrophage Progenitor (GMP: cKit+, CD150+, CD41− FcGRII/III+), megakaryocyte progenitor (MP: cKit+, CD150+, CD41+), erythrocyte progenitor (EP: cKit+, CD150+, FcGRII/III−, Endoglin+), granulocyte/macrophage (GM; Ter119−, CD11b+ Gr1+), Common Lymphoid Progenitor (CLP: Lin−, IL7R+, Flk2+, Sca1$^{mid}$, cKit$^{mid}$), B-cell (Ter119−, CD11b−, Gr1−, B220+), T-cell (Ter119−, CD11b−, Gr1−, CD3+), Platelet (FSC$^{lo}$, Ter119−, CD61+). Gates were defined by n-1 FMO.

### Proliferation of HSCs and MPPs

Fetal liver cells were processed into a single-cell suspension and cKit-enriched using CD117 MicroBeads (Miltenyi Biotec, San Diego, CA, USA). The cKit-enriched population was stained with an antibody cocktail for surface markers of hematopoietic stem and progenitor cells. Cells were then fixed and permeabilized with the True-Nuclear Transcription buffer set (Biolegend) and then stained with Ki67-APC (Invitrogen, Carlsbad, CA, USA) and Hoescht 33,342 (Invitrogen).

### Transplantation assays

Transplantation assays were performed as recently described (López et al, 2022). Briefly, FlkSwitch mice were used as donors for cell isolation and 8- to 12-week-old WT C57BL/6 were used as recipients. Sex of recipients was random and split evenly between male and female.

GFP+ drHSCs and Tom+ HSCs were sorted from fetal liver. Recipient C57BL/6 mice (8–12 weeks) were lethally irradiated using 1,000 cGy (split dose, Precision X-Rad 320). $5 \times 10^5$ whole bone marrow cells from untreated age-matched C57BL/6 and 200 sorted GFP+ drHSCs or Tom+ HSCs wells were diluted in PBS and transplanted via retro-orbital injection using a 1 ml tuberculin syringe in a volume of 100–200 µl. Peripheral blood chimerism was determined in recipients by blood collection via cheek bleeds every 4 weeks for 16 weeks and cells were analyzed by flow cytometry using the LSRII (BD Biosciences). Long-term multilineage reconstitution (LTMR) was defined as chimerism ≥ 1% for each mature blood lineage in the primary transplant and ≥ 0.1% in secondary recipients. At 18 weeks, recipients were euthanized, and BM populations were assessed for chimerism by flow cytometry. $5 \times 10^6$ WBM from fully reconstituted Tom+ HSC primary recipients or $1 \times 10^7$ WBM from GFP+ drHSC primary recipients were transplanted into lethally irradiated secondary recipients. Secondary transplant recipients were monitored for 16 weeks and assessed for peripheral blood output of mature blood cell populations every 4 weeks. At 18 weeks, bone marrow chimerism was assessed by flow cytometry.

### Quantitation of cytokines in fetal amniotic fluid and liver

Fetal amniotic fluid was collected from amniotic sacs using a 1-ml tuberculin syringe. Fetuses were dissected, and individual fetal livers were homogenized and pelleted to collect supernatant. Supernatant was frozen at −80°C and samples were analyzed on an LSRII (BD Biosciences) following the manufacturer's recommendations using a custom LEGENDplex™ (Biolegend) bead-based immunoassay for the following cytokines: IL-23, IL-1α, IFNγ, TNF-α, IL-6, IL-1β, IL-10, IL-17A, IL-27, IFNβ, IFNα, and GM-CSF. Data were analyzed with LEGENDplex™ online analysis platform (https://legendplex.qognit.com/).

## Data availability

This study includes no data deposited in external repositories.

**Expanded View** for this article is available online.

## Acknowledgements

We thank Drs. Angel Kongsomboonvech, Scott Souza for assistance with parasite passaging. We thank Dr. David Gravano at UC Merced's Stem Cell Instrumentation Foundry (SCIF) for flow cytometry support and James Marvin and University of Utah Flow Cytometry Core. KJ was funded by grants AI137126 and AI145403. This work was supported by an NIH/NHLBI award K01HL130753 to AEB, the Pew Biomedical Scholars award to AEB, and the Hellman Fellows Award to AEB. DAL is supported by NIH/NICHD T32HD007491. KSO is supported by NIH/NIAID T32 AI138954.

## Author contributions

**Diego A López:** Conceptualization; data curation; formal analysis; supervision; investigation; visualization; methodology; writing – original draft; project administration; writing – review and editing. **Kelly S Otsuka:** Conceptualization; data curation; formal analysis; investigation; visualization; methodology; writing – original draft; writing – review and editing. **April C Apostol:** Conceptualization; investigation; methodology; writing – original draft. **Jasmine Posada:** Investigation. **Juan C Sánchez-Arcila:** Investigation. **Kirk DC Jensen:** Conceptualization; writing – original draft; writing – review and editing. **Anna E Beaudin:** Conceptualization; formal analysis; supervision; funding acquisition; visualization; methodology; writing – original draft; project administration; writing – review and editing.

## Disclosure and competing interests statement
The authors declare that they have no conflict of interest.

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
