## [Review Process File · The EMBO Journal]

Both maternal IFN γ exposure and acute prenatal infection with *Toxoplasma gondii* activate fetal hematopoietic stem cells

Diego López, Kelly Otsuka, April Apostol, Jasmine Posada, Juan Sánchez-Arcila, Kirk Jensen, and Anna Beaudin
DOI: [10.15252/emboj.2022112693](https://doi.org/10.15252/emboj.2022112693)

Corresponding author(s): Anna Beaudin (anna.beaudin@hsc.utah.edu)

Review Timeline:

Submission Date:	26th Sep 22
Editorial Decision:	15th Nov 22
Revision Received:	10th Feb 23
Editorial Decision:	11th Mar 23
Revision Received:	5th Apr 23
Editorial Decision:	10th Apr 23
Revision Received:	14th Apr 23
Accepted:	28th Apr 23

Editor: Kelly Anderson

Transaction Report:

Dear Dr. Beaudin,

Thank you for submitting your manuscript for consideration by the EMBO Journal. It has now been seen by three referees whose comments are shown below.

Given the referees' positive recommendations, I would like to invite you to submit a revised version of the manuscript, addressing the comments of all three reviewers. I should add that it is EMBO Journal policy to allow only a single round of revision, and acceptance of your manuscript will therefore depend on the completeness of your responses in this revised version. It would be good to discuss your plan to address the referee concerns and I am available to do so by email or zoom in the coming weeks, please email me to schedule a convenient time.

We generally allow three months as standard revision time. As a matter of policy, competing manuscripts published during this period will not negatively impact on our assessment of the conceptual advance presented by your study. However, we request that you contact the editor as soon as possible upon publication of any related work, to discuss how to proceed. Should you foresee a problem in meeting this three-month deadline, please let us know in advance and we may be able to grant an extension. I have attached a guide for revisions to this email, for more information please visit our website.

Thank you for the opportunity to consider your work for publication. I look forward to your revision.

Yours sincerely,

Kelly M Anderson, PhD
Editor
The EMBO Journal
k.anderson@embojournal.org

We realize that it is difficult to revise to a specific deadline. In the interest of protecting the conceptual advance provided by the work, we recommend a revision within 3 months (13th Feb 2023). Please discuss the revision progress ahead of this time with the editor if you require more time to complete the revisions. Use the link below to submit your revision:

Referee #1:

Overall Summary:

The manuscript studied the mechanisms by which congenital toxoplasma infection affects fetal HSC development. Specifically, the manuscript intended to demonstrate the role of IFN γ as a critical regulator of fetal HSC function in context of congenital toxoplasma infection. Overall, this study is highly relevant to a disease affecting pregnant mothers and their infants. The findings are novel and will potentially draw attention from researchers spanning the field of developmental origin of HSC functions, immune regulations etc. There are numerous weaknesses noted. The main weakness can be addressed by having an experiment that assess HSC function using IFN γ knockout mice in setting of maternal toxoplasma infection to support the direct contribution of IFN γ as a regulator of fetal HSC function during maternal toxoplasma infection.

Major Concerns

1) The results presented demonstrated the maternal toxo infection induces cytokine release in dams and fetus (Fig 3), and results in alterations in fetal HSC function (Fig 1, 2). The hypothesis, as written, was "...that fetal HSCs may be directly responsive to IFN γ during infection". While the results presented in Fig 4-6 are required to inform the direct effect of IFN γ , the direct evidence testing the role of IFN γ during maternal toxo infection is lacking. An experiment assessing fetal HSC function using IFN γ knockout mice in setting of maternal toxoplasma infection should be able to address this.

2) To further elaborate on point 1 - the maternal serum shows higher IFN γ in less severe infection (Pru, approx. 50,000 pg/mL vs <10,000pg/mL in RH), while in fetal amniotic fluid there seems to be a dose-response increase in IFN γ according to the virulence of Toxo strain. The IFN γ injection experiments presented, however, achieved maternal serum levels that are approx.

500 pg/mL in IFN γ -/- mice, and 1500 pg/mL in IFN γ -/- mice. This is an understandable weakness of any cytokine treatment, and again stresses the need of having an experiment recommended in point 1.

3) One theme highlighted by authors were the virulence of RH strain (LD100<10) - were there any maternal deaths with 2X10⁴ tachyzoites injection?

4) Gating strategy is typically the same across all experiments, but the "gate" shown in Fig 1C-E appears different (ie gate for HSPC looks different in Ctrl, Pru, and RH). Can authors clarify why the gating appears different?

5) Should the trend of changes in LT-HSC and ST-HSC in response to experiments correlate with Tom⁺ and GFP⁺ cells (e.g Fig 1G-J)? If so, can authors explain the reason behind the discrepancy in Fig 4 B-E?

6) There is a need to include n of both dams/litters and n of pups in figure legends (missing in Fig 1, Fig 2, Fig 4, Fig 6 legends).

7) Methods: There are some discrepancies/details lacking

a. There are some discrepancies in the transplantation assay description. Within text it was described as 5 X 10⁵ (this sounds right), but the cited biorxiv article (Lopez 2022) was 5X10⁶, and the figure 2A schematic is 5 X 10⁴. Please double check.

b. The dose of IFN γ dose was briefly mentioned to be 20ug. This should be included within the methods, along with the origin/vendors of the IFN γ , reasoning of using this dose (is it to achieve the same levels as maternal toxo infection?), and explanation as to how this is reflective/different from an E10.5 infection is important.

c. Collection of amniotic fluid - were steps taken to avoid contamination from residual IFN γ from IP injections?

Minor Comments

1) Few minor clarifications for experimental methodology. For flow cytometry - was isotype control used as a guide to gating?

Transplantation assay: was there any animal death post-transplant?

2) Results: Typically, the number of ST-HSC is higher than LT-HSC in numbers (e.g PMID:34818550), but in Fig 1G vs 1H and Fig 4B vs. 4E, the ST- and LT-HSC appear to be very similar in total cell number.

3) sFlg4 A and C - the IFN γ on top of graphs - is that meant to be "saline"?

4) It will be helpful for readers to have a table similar to supplemental Fig4F to summarize all the findings across infection, IFN injection, and IFN injection with KO.

5) There is a typographical error within the acknowledgement "NHBLI  NHLBI"

6) Within discussion, author stated "transplantation of cells infected with even one RH parasite would have resulted in immediate death" with citation of Mordue et al 2001, which used 100 RH. Did author mean PMID: 8945565? There is another paper that is more relevant that evaluated impact of toxoplasma infection in context of BMT (e.g PMID 31681783).

Referee #3:

The paper "Fetal hematopoietic stem cells are activated by IFN γ during acute prenatal infection with *Toxoplasma gondii*" presents an innovative angle of the study of the impact of inflammation during pregnancy on the fetus - the development of the hematopoietic system.

Maternal inflammation during pregnancy was shown to be harmful to several fetal systems, and to modify the fetal hematopoietic system. This holds true for both vertically transmitted pathogens and those that infect only the mother.

The authors studied a model system that mimics the non-vertical transmitted pathogens by inducing maternal inflammation with Poly-IC (published as a preprint, Lopez et al), and here they added the complimentary study of a vertically transmitted pathogen, *Toxoplasma gondii*.

This is an important extension of the published work because it is still not clear if vertically transmitted pathogens harm the fetus through a different mechanism than those that are restricted to the mother.

Further, the importance of the focus on the hematopoietic system lies in the association between maternal inflammation and the development or "training" of fetal immunity. Long-term impact on the immune or hematologic functions of the fetus can change the quality of life of the adult. Apostol et al studied the long-term function of the fetal hematopoietic stem and progenitor cells by harnessing the FlkSwitch system (Beaudin et al 2014). This is an adequate tool that offers a unique perspective on the possible impact of maternal inflammation on fetal development. Moreover, this paper nicely addresses the need to isolate the effect of IFN γ from the broad effect of the pathogen, *Toxoplasma gondii*, on the fetal hematopoietic system.

Therefore, I find the topic, the approach, and the results very important and well done. I also find the paper well-written and easy to read and follow. There are some points that a better introduction or explanation will make the paper more accessible to a wider reader audience, and I comment on that below. Other addressable points are listed below as well.

Major

1. One of the main conclusions of the paper is "Thus, severity of maternal infection appeared to drive immediate virulence dependent changes to proliferation by triggering HSC proliferation and expansion of downstream HSPCs in the fetal liver" but is not supported by the data in its current form: Given that two populations (GFP+drHSC and Tom+HSC) showed decreased cell number following infection with either strain of Toxo (Figure 1 I and J) but increased proliferation by Ki67 staining in both strains (Figure 1 N and O), it is likely that cell death is an important driver of the phenotype. Because the authors didn't address cell death in the figure, it is hard to evaluate the contribution of that factor and to exclude the option that the increased proliferation is a compensatory response to cell death caused by the pathogen, or by IFN γ . In this case, the changes in proliferation are a secondary effect and the conclusion above should be modified, or cell death should be addressed experimentally.

2. Please better explain the HSC and HSPC lineage profiling so it will be accessible to a broader audience. Most importantly, the

developmentally-restricted HSCs that were discovered and characterized by the authors (Beaudin et al 2014) require explanation, and a specific clarification for their relevance in the current study. Please explain the importance of this population in the fetus and the impact of its perturbation for the adult immune and hematologic functions in the introduction (the discussion section contains an explanation, but it is important to introduce the biological question upfront). This will emphasize the impact of the work presented here. Additionally, a visual presentation such as a lineage map of all the subpopulations presented in Figure 1 (and throughout the paper) will be very helpful both when the system is first presented and as an accompanied scheme for data in Figure 2.

3. Data presentation can be improved in terms of consistency between figures. This includes the following:

3.1 Add graphs with p values to 1C-E

3.2 Figure 3 should present the data consistently between amniotic fluid, fetal liver, and maternal serum: all three should have the graphs for the same cytokines in the same order, and - most importantly - a heat map for the cytokines in the maternal serum should be included in the main figure. This is important for easy comparison between these three sites.

3.3 Given the focus on IFN γ later in the manuscript, please show IFN γ data in clear graphs in Figure 3. Specifically, this is important to be shown in fetal liver, since the authors indicate the possible impact of IFN γ on HSCC and HSPC in their environment, which is, in the fetus's case, the fetal liver.

4. It is interesting to consider whether the Pru and RH strains also differ in their vertical transmission. This can help in understanding the different cytokine profiles between these two strains and in drawing conclusions about the similarity between Pru infection and IFN γ treatment. Is it possible to test this?

5. Please explain the reasoning behind the choice of the focus on the presented cytokines. For example, why IL-1 α is presented in the paper while the MIA literature often focuses on IL-6, IL-10, IL-1 β , CCL2, TNF α and IFN γ .

6. In the literature, IFN γ levels in MIA mouse models and in patients with inflammation usually less than 10 pg/ml, and often measured around 1-2 pg/ml. This range is close to what the authors observed in Figure S4C. However - in Figure 3I, Toxo infection in dams resulted in $n \times 10^4$ pg/ml IFN γ . In other graphs, other, quite diverse levels were measured: in Figure S4 in dams injected with IFN γ the levels go up to 500 pg/ml in the IFN γ R $^{+/-}$ dams; in the fetus IFN γ levels reach a maximum of 15 pg/ml in the liver, and <200 pg/ml in the amniotic fluid. The differences between detected levels of IFN γ are very big and raise the question about the functional significance of some of the changes (the statistical significance is there but does not address this point), or alternatively - a reason to double check that the units are correct in all graphs.

Minor

1. Figure 1P - What is the meaning of the changes in Ki67 from 90% to 95%? Is there a functional difference between these high numbers?

2. Figure SFig.1B: Please include in the legend a clarification regarding the statistics. Is it that 50% of each litter lived or 50% of all litters combined (so it is possible that one litter had 100% death while another had 0%?). Also - Do you know what day the fetuses die? If yes - please mention this.

3. In the section "In utero exposure to IFN γ activates fetal hematopoiesis", it will be helpful to remind the reader why IFN γ was chosen for the follow-up work. This is nicely presented in the intro but not the focus of Figure 3 so the flow feels broken here but can be nicely stitched together with a reminder of the biological question and known impact of IFN γ on fetal development, independent of Toxo, and in the context of Toxo, independent of embryonic development. This might be less of an issue if Figure 3 will include a good presentation of IFN γ data as suggested above.

Additional suggestions

1. Move figures SFig.1D and SFig1E to "main". These are important for the understanding of 1F-M.

2. Please add an explanation for the result for the Pru cohort in Figure 2K, as opposed to this cohort in 2P-R: what is the explanation of the higher chimerism in the 2 successful recipients? Is this indicative of some competition or threshold in the secondary transplantation?

3. The titles of Figures 4 and 5, and the parallel sections in the text are based on a conclusion prematurely drawn: we learn that "in utero exposure to IFN γ " is what activates fetal hematopoiesis only later in the manuscript. At this stage of the story, we are still not sure if it is maternal or fetal exposure that drives the phenotype because data in Figure 6 addresses this point. It is worth considering replacing the titles so they fit the level of knowledge as it unfolds when reading the paper.

4. For the experiment shown in Figure 4 - show that the injected IFN γ reached the fetus. This is important to support the hypothesis that IFN γ influenced fetal hematopoiesis through changes in the hematopoietic environment. This is only a suggestion because this point is thoroughly addressed later, and therefore not critical for the conclusions here but will make Figure 4 far better.

5. The section that begins with "To further analyze the fetal HSC response to IFN γ ,..." is very important because it addresses the critical point of distinguishing between the primary and secondary effect of IFN γ that is injected to dams. I suggest moving the data from SFig6F to the main figure and enhance the discussion of the results to emphasize that point.

Response to Reviewers

Please find our responses to reviewer *in italics* below original reviewer comments.

Referee #1:

Overall Summary:

The manuscript studied the mechanisms by which congenital toxoplasma infection affects fetal HSC development. Specifically, the manuscript intended to demonstrate the role of IFN γ as a critical regulator of fetal HSC function in context of congenital toxoplasma infection. Overall, this study is highly relevant to a disease affecting pregnant mothers and their infants. The findings are novel and will potentially draw attention from researchers spanning the field of developmental origin of HSC functions, immune regulations etc. There are numerous weaknesses noted. The main weakness can be addressed by having an experiment that assess HSC function using IFN γ knockout mice in setting of maternal toxoplasma infection to support the direct contribution of IFN γ as a regulator of fetal HSC function during maternal toxoplasma infection.

Major Concerns

1) The results presented demonstrated the maternal toxo infection induces cytokine release in dams and fetus (Fig 3), and results in alterations in fetal HSC function (Fig 1, 2). The hypothesis, as written, was "...that fetal HSCs may be directly responsive to IFN γ during infection". While the results presented in Fig 4-6 are required to inform the direct effect of IFN γ , the direct evidence testing the role of IFN γ during maternal toxo infection is lacking. An experiment assessing fetal HSC function using IFN γ knockout mice in setting of maternal toxoplasma infection should be able to address this.

We agree with the reviewer that these suggested experiments could potentially address the direct role of IFN γ in mediating the effects of toxoplasma infection on HSC function. However, we feel strongly that this experiment is beyond the scope of the current manuscript for several technical reasons. One, the degree of vertical transmission that occurs in the complete absence of maternal IFN γ in the IFN γ knockout mice will most certainly affect the interpretation of any findings from these experiments. We have instead decided to change some of the language throughout the manuscript, including our title, to indicate that we are independently studying the effects of two separate maternal stimuli IFN γ AND toxoplasmosis, as opposed to IFN γ as the only mediator of the effects of toxoplasma during pregnancy.

*We would like to mention that we have also performed experiments in which we infected pregnant IFN γ -/- dams on three separate occasions with Toxoplasma in our model, and only once did the dams survive (barely) to E15.5. At this time point, the fetal units were extremely compromised, with drastic losses in fetal liver cellularity making interpretation problematic. From this, we decided to explore the impact of IFN γ , as this cytokine is central to host defense to *T. gondii* and a critical regulator of HSC function.*

2) To further elaborate on point 1 - the maternal serum shows higher IFN γ in less severe infection (Pru, approx. 50,000 pg/mL vs <10,000pg/mL in RH), while in fetal amniotic fluid there seems to be a dose-response increase in IFN γ according to the virulence of Toxo strain. The IFN γ injection experiments presented, however, achieved maternal serum levels that are approx. 500 pg/mL in IFN+/- mice, and 1500 pg/mL in IFN-/- mice. This is an understandable

weakness of any cytokine treatment, and again stresses the need of having an experiment recommended in point 1.

We agree with the reviewer, except we would like to note that differences in IFN γ levels within the fetal liver niche are not nearly as stark (Fig. 3K and SFig 4D). We consider these differences to be more relevant to our story, and to HSC development at this timepoint, and this is reflected in comparable responses of HSCs in response to infection and IFN γ injection.

3) One theme highlighted by authors were the virulence of RH strain (LD100<10) - were there any maternal deaths with 2×10^4 tachyzoites injection?

Yes, moms died before birth, which is why we investigated hematopoiesis at a prenatal timepoint. This is another reason that the suggested experiments in IFN γ KO mice proved to be very difficult.

4) Gating strategy is typically the same across all experiments, but the "gate" shown in Fig 1C-E appears different (ie gate for HSPC looks different in Ctrl, Pru, and RH). Can authors clarify why the gating appears different?

We very much appreciate the reviewer pointing this out, as it prompted us to review gating across all experiments. We found significant scaling errors in analysis performed by a former PhD student who had remained in California when our lab moved to the University of Utah at the start of the pandemic. That student has since left the lab and left science. The two co-first authors have subsequently reanalyzed all data and/or performed additional experiments as needed throughout the manuscript. This re-analysis clarified much of our data, including unexplained differences in cellularity between Tom+ and GFP+ HSCs vs LT- and ST- HSCs (see response to point #5, below), and gating differences between groups, without significant changes in the overall outcomes of the study. We have therefore now included revised representative FACS plots that reflect these revised analyses with improved scaling, and included updated graphs for cellularity/frequency that reflected these revised analyses.

5) Should the trend of changes in LT-HSC and ST-HSC in response to experiments correlate with Tom+ and GFP+ cells (e.g Fig 1G-J)? If so, can authors explain the reason behind the discrepancy in in Fig 4 B-E?

We define Tom+ and GFP+ HSCs using a CD150+ LSK gate (see SFig 1D), whereas LT-HSCs and ST-HSCs are gated as conventional HSCs defined by Flk2/CD48/CD150 LSK. Thus, technically Tom+ and GFP+ HSCs fall into a gate that includes some phenotypic MPP2s, although backgating usually places Tom+ HSCs closer to phenotypic LT-HSCs, whereas GFP+ HSCs can overlap more with phenotypic MPP2s. This trend can now be observed in Fig. 1, where Tom+ HSCs (Fig. 1I) follow a trend similar to LT-HSCs (Fig. 1G) and GFP+ HSCs (Fig. 1J) follow a trend similar to MPP2s (Fig. 1K).

6) There is a need to include n of both dams/litters and n of pups in figure legends (missing in Fig 1, Fig 2, Fig 4, Fig 6 legends).

We thank the reviewer for pointing out this omission and have added this information to all figure legends.

7) Methods: There are some discrepancies/details lacking

a. There are some discrepancies in the transplantation assay description. Within text it was described as 5×10^5 (this sounds right), but the cited biorxiv article (Lopez 2022) was 5×10^6 , and the figure 2A schematic is 5×10^4 . Please double check.

We appreciate the reviewer catching this discrepancy and have corrected the text in the manuscript to be 5×10^5 WBM competitors in the primary transplant and 5×10^6 WBM cells for the secondary transplant.

b. The dose of IFN γ dose was briefly mentioned to be 20ug. This should be included within the methods, along with the origin/vendors of the IFN γ , reasoning of using this dose (is it to achieve the same levels as maternal toxo infection?), and explanation as to how this is reflective/different from an E10.5 infection is important.

We thank the reviewer for bring this point to our attention. We have now included the dose in the methods (Mouse models and husbandry), along with the vendor. The dose chosen was based on previous studies using 1-2 doses of 10ug as well as preliminary experiments in pregnant mice aimed at modeling similar levels of IFN γ in the fetal liver as compared to infection experiments. As the reviewer acknowledges, even comparing cytokine levels does not appropriate model an evolving infection with a single dose of IFN γ .

c. Collection of amniotic fluid - were steps taken to avoid contamination from residual IFN γ from IP injections?

Yes, the entire uterine horn was removed and rinsed off before amniotic fluid was collected directly from each embryo after it was removed from the uterine horn.

Minor Comments

1) Few minor clarifications for experimental methodology. For flow cytometry - was isotype control used as a guide to gating? Transplantation assay: was there any animal death post-transplant?

We use Fluorescence Minus One (FMO) controls to determine proper gate placement (updated within the methods section). No animal death from parasite was observed post-transplant.

2) Results: Typically, the number of ST-HSC is higher than LT-HSC in numbers (e.g PMID:34818550), but in Fig 1G vs 1H and Fig 4B vs. 4E, the ST- and LT-HSC appear to be very similar in total cell number.

This is not the case in fetal hematopoiesis when MPPs are gated conventionally (e.g. Challen et al, 2021; PMID: 34648848), and can be seen in frequencies depicted in Fig. 1C.

3) sFlg4 A and C - the IFN γ on top of graphs - is that meant to be "saline"?

Thank you for pointing this out, we have corrected the labels.

4) It will be helpful for readers to have a table similar to supplemental Fig4F to summarize all the findings across infection, IFN injection, and IFN injection with KO.

If this is absolutely necessary we can include it, but we found assembling a table of this type across each infection and injection condition cumbersome and somewhat repetitive.

5) There is a typographical error within the acknowledgement "NHBLI  NHLBI"

We thank the reviewer for catching this error and have corrected this typo.

6) Within discussion, author stated "transplantation of cells infected with even one RH parasite would have resulted in immediate death" with citation of Mordue et al 2001, which used 100 RH. Did author mean PMID: 8945565? There is another paper that is more relevant that evaluated impact of toxoplasma infection in context of BMT (e.g PMID 31681783).

We thank the reviewer for catching this error and have fixed the reference as suggested.

Referee #3:

The paper "Fetal hematopoietic stem cells are activated by IFN γ during acute prenatal infection with *Toxoplasma gondii*" presents an innovative angle of the study of the impact of inflammation during pregnancy on the fetus - the development of the hematopoietic system.

Maternal inflammation during pregnancy was shown to be harmful to several fetal systems, and to modify the fetal hematopoietic system. This holds true for both vertically transmitted pathogens and those that infect only the mother.

The authors studied a model system that mimics the non-vertical transmitted pathogens by inducing maternal inflammation with Poly-IC (published as a preprint, Lopez et al), and here they added the complimentary study of a vertically transmitted pathogen, *Toxoplasma gondii*. This is an important extension of the published work because it is still not clear if vertically transmitted pathogens harm the fetus through a different mechanism than those that are restricted to the mother.

Further, the importance of the focus on the hematopoietic system lies in the association between maternal inflammation and the development or "training" of fetal immunity. Long-term impact on the immune or hematologic functions of the fetus can change the quality of life of the adult. Apostol et al studied the long-term function of the fetal hematopoietic stem and progenitor cells by harnessing the FlkSwitch system (Beaudin et al 2014). This is an adequate tool that offers a unique perspective on the possible impact of maternal inflammation on fetal development. Moreover, this paper nicely addresses the need to isolate the effect of IFN γ from the broad effect of the pathogen, *Toxoplasma gondii*, on the fetal hematopoietic system.

Therefore, I find the topic, the approach, and the results very important and well done. I also find the paper well-written and easy to read and follow. There are some points that a better introduction or explanation will make the paper more accessible to a wider reader audience, and I comment on that below. Other addressable points are listed below as well.

Major

1. One of the main conclusions of the paper is "Thus, severity of maternal infection appeared to drive immediate virulence dependent changes to proliferation by triggering HSC proliferation and expansion of downstream HSPCs in the fetal liver" but is not supported by the data in its current form: Given that two populations (GFP+drHSC and Tom+HSC) showed decreased cell number following infection with either strain of Toxo (Figure 1 I and J) but increased proliferation by Ki67 staining in both strains (Figure 1 N and O), it is likely that cell death is an important driver of the phenotype. Because the authors didn't address cell death in the figure, it is hard to evaluate the contribution of that factor and to exclude the option that the increased proliferation

is a compensatory response to cell death caused by the pathogen, or by IFN γ . In this case, the changes in proliferation are a secondary effect and the conclusion above should be modified, or cell death should be addressed experimentally.

We understand the reviewer's perspective. Our initial interpretation of these data were not that cells die but rather that they differentiate rapidly in response to inflammatory cues to produce downstream progenitors, as described for adult hematopoiesis (Caiado et al, JEM, 2021). Similar results have been described in response to IFN γ in the context of m. Avium infection, where increased proliferation does not expand HSCs, but does cause an increase in downstream progenitor expansion (Baldrige et al, Nature, 2010). The fact that HSPC cellularity, overall, is maintained or expanded despite dramatic overall decreases in CD45+ cells, supports that assertion. However, we acknowledge that in the case of LT-HSCs and ST-HSCs, we have not directly measured cell death, and cannot exclude that as a cause of decreased cell number following infection. We have therefore modified the text describing Fig. 1 to reflect that conclusion.

2. Please better explain the HSC and HSPC lineage profiling so it will be accessible to a broader audience. Most importantly, the developmentally-restricted HSCs that were discovered and characterized by the authors (Beaudin et al 2014) require explanation, and a specific clarification for their relevance in the current study. Please explain the importance of this population in the fetus and the impact of its perturbation for the adult immune and hematologic functions in the introduction (the discussion section contains an explanation, but it is important to introduce the biological question upfront). This will emphasize the impact of the work presented here. Additionally, a visual presentation such as a lineage map of all the subpopulations presented in Figure 1 (and throughout the paper) will be very helpful both when the system is first presented and as an accompanied scheme for data in Figure 2.

We appreciate the reviewer's desire to highlight the impact of our previous work. As suggested, we have now included a sentence in the last paragraph of the introduction describing that work when we describe the FlkSwitch model. We have also now included a lineage map of the populations that includes our two fate-mapped HSC populations in SFig. 1C as suggested by the reviewer.

3. Data presentation can be improved in terms of consistency between figures. This includes the following:

3.1 Add graphs with p values to 1C-E

We thank the reviewer for this suggestion and have now included graphs depicting all values for frequencies in Sfig. 1F-L.

3.2 Figure 3 should present the data consistently between amniotic fluid, fetal liver, and maternal serum: all three should have the graphs for the same cytokines in the same order, and - most importantly - a heat map for the cytokines in the maternal serum should be included in the main figure. This is important for easy comparison between these three sites.

We appreciate the reviewers suggestion and have now significantly modified Fig. 3 to include all cytokines that are significantly altered in across all three tissues measured. We have also now included a heat map of cytokines for maternal serum in Fig. 3D' and 3E'. We have also rearranged the order of presentation so that all cytokines are presented in the same order across all three tissues.

3.3 Given the focus on IFN γ later in the manuscript, please show IFN γ data in clear graphs in Figure 3. Specifically, this is important to be shown in fetal liver, since the authors indicate the possible impact of IFN γ on HSCC and HSPC in their environment, which is, in the fetus's case, the fetal liver.

We have added these data to the manuscript in Figure 3K, and we thank for reviewer for this suggestion as the comparison of FL IFN γ values has added significant value to the manuscript.

4. It interesting to consider whether the Pru and RH strains also differ in their vertical transmission. This can help in understanding the different cytokine profiles between these two strains and in drawing conclusions about the similarity between Pru infection and IFN γ treatment. Is it possible to test this?

We appreciate the reviewer's interest, but unfortunately this is no longer possible to test as these experiments were completed several years ago in a collaborator's lab at another institution, and the student responsible for these experiments has since left that institution. We believe that vertical transmission is likely different between two strains based on viability outcomes, but we acknowledge in our discussion that this is a limitation of the current study,

5. Please explain the reasoning behind the choice of the focus on the presented cytokines. For example, why IL-1 α is presented in the paper while the MIA literature often focus on IL-6, IL-10, IL-1 β , CCL2, TNF α and IFN γ .

We analyzed a large array of cytokines using a set array and have now included all plots representing cytokine levels for all cytokines measured in our analysis in Figure 3 across amniotic fluid, fetal liver and maternal serum.

6. In the literature, IFN γ levels in MIA mouse models and in patients with inflammation usually less than 10 pg/ml, and often measured around 1-2 pg/ml. This rang is close to what the authors observed in Figure S4C. However - in Figure 3I, Toxo infection in dams resulted in nX104 pg/ml IFN γ . In other graphs, other, quite diverse levels were measured: in Figure S4 in dams injected with IFN γ the levels go up to 500 pg/ml in the IFN γ R \pm dams; in the fetus IFN γ levels reach a maximum of 15 pg/ml in the liver, and <200 pg/ml in the amniotic fluid. The differences between detected levels of IFN γ are very big and raise the question about the functional significance of some of the changes (the statistical significance is there but does not address this point), or alternatively - a reason to double check that the units are correct in all graphs.

We understand the reviewer's surprise, as we also found the range in these data to be of great interest. Note that differences in cytokine levels between tissues (maternal serum, amniotic fluid, fetal liver) are not directly comparable as they reflect very different starting volumes (maternal serum vs fetal liver supernatant, for example). We do not find it surprising that an actual virulent infection produces cytokine levels that far exceed that of a single experimental manipulation (e.g. MIA with poly(I:C)). Note, however, that differences in IFN γ levels within the fetal liver niche (which is most relevant to our story, and to HSC development at this timepoint) are not as stark between Pru infection and a single injection of IFN γ (Fig 3K and SFig 4D), for example, and this is reflected in comparable responses of HSCs under these two conditions.

Minor

1. Figure 1P - What is the meaning of the changes in Ki67 from 90% to 95%? Is there a functional difference between these high numbers?

We understand the reviewer's concern. Yet, small differences in proliferation can have significant effects on cellularity, as observed, for example for MPPs in situ (Fir 1K-M).

2. Figure SFig.1B: Please include in the legend a clarification regarding the statistics. Is it that 50% of each litter lived or 50% of all litters combined (so it is possible that one litter had 100% death while another had 0%?). Also - Do you know what day the fetuses die? If yes - please mention this.

We thank the reviewer for requesting this clarification. Each bar represents the mean percentage of live pups per litter/per condition. In our case, the reported mean closely represents the percentage of live pups per individual litter; we did not, for example, observe 0% viable pups in one litter and 100% pups in another litter, in any infected litter. We have fixed the Y-axis to clarify.

3. In the section "In utero exposure to IFN γ activates fetal hematopoiesis", it will be helpful to remind the reader why IFN γ was chosen for the follow-up work. This is nicely presented in the intro but not the focus of Figure 3 so the flow feels broken here but can be nicely stitched together with a reminder of the biological question and known impact of IFN γ on fetal development, independent of Toxo, and in the context of Toxo, independent of embryonic development. This might be less of an issue if Figure 3 will include a good presentation of IFN γ data as suggested above.

We appreciate the reviewer's suggestion, and have both addressed the issue with Figure 3, referenced that figure, and also included an additional sentence, including two references, on why we chose to focus on IFN γ .

Additional suggestions

1. Move figures SFig.1D and SFig1E to "main". These are important for the understanding of 1F-M.

We appreciate the reviewer's suggestion but feel that Fig. 1 is already very crowded. We have made sure to include it as a supplementary figure, and these data are referred only once in the text of the manuscript.

2. Please add an explanation for the result for the Pru cohort in Figure 2K, as opposed to this cohort in 2P-R: what is the explanation of the higher chimerism in the 2 successful recipients? Is this indicative of some competition or threshold in the secondary transplantation?

We believe that this finding reflects the differential capacity of distinct fetal HSCs to respond to inflammatory mediators induced by congenital infection of varying virulence. We discuss the same conclusion within our discussion section.

3. The titles of Figures 4 and 5, and the parallel sections in the text are based on a conclusion prematurely drawn: we learn that "in utero exposure to IFN γ " is what activates fetal hematopoiesis only later in the manuscript. At this stage of the story, we are still not sure if it is maternal or fetal exposure that drives the phenotype because data in Figure 6 addresses this point. It is worth considering replacing the titles so they fit the level of knowledge as it unfolds when reading the paper.

We appreciate the reviewer's suggestion and have replaced "in utero" with "prenatal".

4. For the experiment shown in Figure 4 - show that the injected IFN γ reached the fetus. This is important to support the hypothesis that IFN γ influenced fetal hematopoiesis through changes in the hematopoietic environment. This is only a suggestion because this point is thoroughly addressed later, and therefore not critical for the conclusions here but will make Figure 4 far better.

We agree that as we demonstrate this later in the paper, we did not feel it was necessary to include it at this stage of the manuscript.

5. The section that begins with "To further analyze the fetal HSC response to IFN γ ,..." is very important because it addresses the critical point of distinguishing between the primary and secondary effect of IFN γ that is injected to dams. I suggest moving the data from SFig6F to the main figure and enhance the discussion of the results to emphasize that point.

We thank the reviewer for the suggestion, however we did not feel moving this data enhances the overall interpretation of the figure as it was only intended to help interpret the cytokine data presented in Sfig 4. If this is absolutely necessary we can include it.

Dear Anna,

Congratulations on a great revision! Overall, the referees have been positive and are in support of publication. However, referee 3 has asked you to add some information to Figure 3. When you submit your revised version, please also take care of the following editorial items and add this also to your point-by-point response:

1. Please add up to five keywords, which may or may not appear in the title, in alphabetical order, below the abstract, with each word separated by a slash.
2. Please add a data availability section. If your manuscript does not contain any datasets, please use the following statement: This study includes no data deposited in external repositories.
3. Please review our new policy on conflict of interests on the EMBO author guide website and update the title of this section to: Disclosure and competing interests statement.
4. Please remove the author contribution section from the main manuscript.
5. We do not allow "data not shown" in our publications. Please remove the reference to this data (pg. 10) or add the data to a figure and update text and legends.
6. We require that all figures be referred to in the main text and in alphabetical order. Please double check that all figures and supplementary figures are referred to and in the correct order. (Figure 1O not referenced, Figure 2, 4, 5, 6 have figures called out of order).
7. Please add the manuscript number to the author checklist
8. Please upload the point-by-point response separately, rather than as part of the cover letter as the point-by-point response will be publicly available and the cover letter is not public.
9. Please upload the main figures as individual, high-resolution figure files and remove them from the manuscript text. Up to five EV figures (from the supplemental file) can also be uploaded as individual figure files and their legends should then be added to the manuscript, after the main figure legends.
10. Supplementary figures should be called out either as EV figures with legends included in the manuscript file, and renamed to Figure EV1-EV4, etc. or included in the Appendix PDF with the legends and renamed as Appendix Figure S1-S4, etc.
11. We encourage the publication of source data, particularly for electrophoretic gels and blots and graphs, with the aim of making primary data more accessible and transparent to the reader. It would be great if you could provide me with a PDF file per figure that contains the original, uncropped and unprocessed scans of all or key gels used in the figure or for graphs, an Excel spreadsheet with the original data used to generate the graphs. The PDF files should be labeled with the appropriate figure/panel number, and should have molecular weight marker; further annotation could be useful but is not essential. The PDF files will be published online with the article as supplementary "Source Data" files.
12. We include a synopsis of the paper (see <http://emboj.embojpress.org/>). Please provide me with a general summary statement and 3-5 bullet points that capture the key findings of the paper.
13. We also need a summary figure for the synopsis. The size should be 550 wide by 200-440 high (pixels). You can also use something from the figures if that is easier.

Thank you for the opportunity to consider your work for publication. I look forward to your revision.

Kind regards,
Kelly

Kelly M Anderson, PhD
Editor
The EMBO Journal
k.anderson@embojournal.org

Further information is available in our Guide For Authors: <https://www.embojpress.org/page/journal/14602075/authorguide>

Use the link below to submit your revision:

Referee #1:

The revised manuscript is much stronger and clearer. I do not have any major comment. One very minor non-essential comment would be to omit the summary of the study result from the last paragraph of introduction. again, this is not essential but more of a presentation style.

One final comment that does not need to be addressed: One factor to strongly consider for future studies is to assess impact of infection on maternal weight gain - seems like the dams infected by the RH strain is really not well, and that could affect diet intake etc resulting in smaller fetuses/smaller liver/liver with lesser cells. This makes fetal programming researchers wonder if the findings relate to maternal undernutrition/fetal IUGR. It would be important and interesting data point assess in the future studies.

Referee #3:

The authors addressed my concerns, but should add titles and units to the graphs in Figure 3 before publication.
Congratulations!

We have made all suggested changes, including the addition of the final reviewer comment to the end of the response to reviewers. Please let us know if any additional changes are needed. We are very much looking forward to seeing our work published in EMBO Journal.

Dear Anna,

Thank you for providing a revised version of your manuscript. There remains however two editorial issues to attend to before we can move forward. When you submit your revised version, please add the following to your new point-by-point response:

1. Please rename EV figures to Figure EV1-EV4 rather than Appendix Figure S1-S4, with the correct callouts in the manuscript file.
2. We strongly advise you to upload source data, particularly for electrophoretic gels and blots and graphs, with the aim of making primary data more accessible and transparent to the reader. In the near future this will become mandatory for all EMBO journal articles and for many funding agencies. It would be great if you could provide me with a PDF file per figure that contains the original, uncropped and unprocessed scans of all or key gels used in the figure or fo graphs, an Excel spreadsheet with the original data used to generate the graphs. For example Figure 1B, F-O. Figure 2C-J, L-S. Figure 3A-I, K-S, U-C'. Figure 4A-H. Figure 5B-I, K-R. Figure 6 would all have source data appropriate for publication alongside the figures. You should have received an email from Hannah Sonntag with more details.

Thank you for the opportunity to consider your work for publication and I look forward to receiving your revision.

Kind regards,
Kelly

Kelly M Anderson, PhD
Editor
The EMBO Journal
k.anderson@embojournal.org

Use the link below to submit your revision:

All editorial and formatting issues were resolved by the authors.

Dear Anna,

Congratulations on an excellent manuscript, I am pleased to inform you that your manuscript has been accepted for publication in the EMBO Journal. Thank you for your comprehensive response to the referee concerns and for providing detailed source data. It has been a pleasure to work with you to get this to the acceptance stage.

I will begin the final checks on your manuscript before submitting to the publisher next week. Once at the publisher, it will take about three weeks for your manuscript to be published online. As a reminder, the entire review process, including referee concerns, and your point-by-point response will be available to readers.

I will be in touch throughout the final editorial process until publication. In the meantime, I hope you find time to celebrate!

Kind regards,

Kelly

Kelly M Anderson, PhD
Scientific Editor, USA
The EMBO Journal
k.anderson@embojournal.org

Please note that it is EMBO Journal policy for the transcript of the editorial process (containing referee reports and your response letter) to be published as an online supplement to each paper. If you do NOT want this, you will need to inform the Editorial Office via email immediately. More information is available here:
<https://www.embopress.org/page/journal/14602075/authorguide#transparentprocess>

Your manuscript will be processed for publication in the journal by EMBO Press. Manuscripts in the PDF and electronic editions of The EMBO Journal will be copy edited, and you will be provided with page proofs prior to publication. Please note that supplementary information is not included in the proofs.

You will be contacted by Wiley Author Services to complete licensing and payment information. The required 'Page Charges Authorization Form' is available here: https://www.embopress.org/pb-assets/embo-site/tej_apc.pdf - please download and complete the form and return to embopressproduction@wiley.com

EMBO Press participates in many Publish and Read agreements that allow authors to publish Open Access with reduced/no publication charges. Check your eligibility: <https://authorservices.wiley.com/author-resources/Journal-Authors/open-access/affiliation-policies-payments/index.html>

Should you be planning a Press Release on your article, please get in contact with embojournal@wiley.com as early as possible, in order to coordinate publication and release dates.
